# Stationary MMD Points

**Zonghao Chen** [1]   **Toni Karvonen** [2]   **Heishiro Kanagawa** [3]   **François-Xavier Briol** [1]   **Chris J. Oates** [3 4]

## Abstract

Approximation of a target probability distribution using a finite set of points is a problem of fundamental importance in numerical integration. Several authors have proposed to select points by minimising a maximum mean discrepancy (MMD), but the non-convexity of this objective typically precludes global minimisation. Instead, we consider the concept of *stationary points of the MMD* which, in contrast to points globally minimising the MMD, can be accurately computed. Our main contributions are two-fold. We first prove the (perhaps surprising) result that, for integrands in the associated reproducing kernel Hilbert space, the integration error of stationary MMD points vanishes *faster* than the MMD. Motivated by this *super-convergence* property, we consider MMD gradient flows as a practical strategy for computing stationary points of the MMD. We then prove that MMD gradient flow can indeed compute stationary MMD points, based on a refined convergence analysis that establishes a novel non-asymptotic finite-particle error bound.

## 1. Introduction

This paper is concerned with the task of approximating a given probability distribution $\mu$ on $\mathbb{R}^d$ using a finite set of $n$ particles $\{\boldsymbol{x}_i\}_{i=1}^n$, with a view to performing numerical integration (or *cubature*) of a black-box function $f\colon \mathbb{R}^d \to \mathbb{R}$ with respect to $\mu$, which requires careful selection of the nodes at which the integrand $f$ is evaluated [36]. Numerical integration is for example crucial for computing posterior expectations in Bayesian inference [74], but also in optimisation and reinforcement learning, where objective functions

[1]University College London, London, UK [2]Lappeenranta–Lahti University of Technology LUT, Lappeenranta, Finland [3]Newcastle University, Newcastle upon Tyne, UK [4]The Alan Turing Institute, London, UK. Correspondence to: Zonghao Chen <zonghao.chen.XX@ucl.ac.uk>, Chris J. Oates <chris.oates@newcastle.ac.uk>.

*Proceedings of the 43$^{rd}$ International Conference on Machine Learning*, Seoul, South Korea. PMLR 306, 2026. Copyright 2026 by the author(s).

are expressed as expectations under $\mu$ and are approximated via samples [17, 49]. When $\mu$ corresponds to the empirical distribution of a large dataset, this task reduces to selecting a smaller set of representative points (i.e., a *coreset*), which forms an accurate approximation to $\mu$ and can be used to reduce computational cost in downstream tasks [7].

Different strategies for selecting points $\{\boldsymbol{x}_i\}_{i=1}^n$ exist depending on how the accuracy of the approximation is measured. This work considers approximation in the sense of MMD [48]

$$\mathrm{MMD}^2(\mu, \mu_n) = \iint k(\boldsymbol{x}, \boldsymbol{y})\mathrm{d}(\mu - \mu_n)(\boldsymbol{x})\mathrm{d}(\mu - \mu_n)(\boldsymbol{y}),$$

where $\mu_n = \frac{1}{n}\sum_{i=1}^n \delta_{\boldsymbol{x}_i}$. MMD uses a symmetric positive semi-definite *kernel* $k\colon \mathbb{R}^d \times \mathbb{R}^d \to \mathbb{R}$ [15] to measure the discrepancy between $\mu$ and the empirical measure $\mu_n$ of the points $\{\boldsymbol{x}_i\}_{i=1}^n$. This choice is motivated by its generality (a wide range of topologies can be induced via the choice of kernel [90, 11]), practicality (MMD can often be explicitly computed [22]), and favourable (e.g., dimension-independent) sample complexity compared to alternative measures of discrepancy [48]. Further, the MMD is related to numerical integration error for functions $f$ in the reproducing kernel Hilbert space $\mathcal{H}$ (RKHS) associated to the kernel $k$ via

$$\left| \tfrac{1}{n}\sum_{i=1}^n f(\boldsymbol{x}_i) - \int f \mathrm{d}\mu \right| \le \|f\|_{\mathcal{H}} \cdot \mathrm{MMD}(\mu, \mu_n), \quad (1)$$

which follows from reproducing property and Cauchy–Schwarz, and holds for *any* point set $\{\boldsymbol{x}_i\}_{i=1}^n$.

Several works have studied theoretical properties of point sets $\{\boldsymbol{x}_i\}_{i=1}^n$ that minimise MMD, deducing fast convergence rates for numerical integration when $\mu$ is uniform over the unit sphere [20, 75], the unit cube $[0,1]^d$ [36], or has sub-exponential tails [96]. On the practical side, a range of algorithms have been proposed to approximate such minimum MMD points, showing strong empirical performance in numerical integration tasks. Notable examples include sequential greedy methods [26, 93], particle-based minimization schemes [96, 27], and convex–concave procedures [73].

However, a fundamental gap remains between these theoretical results and practical algorithms: due to the non-convexity of the MMD objective, none of the existing algorithms can provably recover the true *minimum* MMD points

| IID Samples | MMD Flow (T=0) | MMD Flow (T=20) | Stationary MMD Points |
|:---:|:---:|:---:|:---:|
| 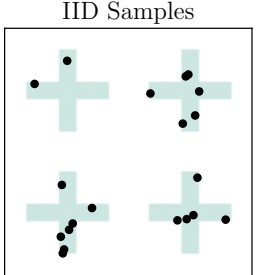 | 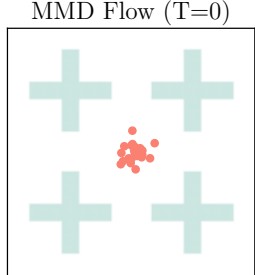 | 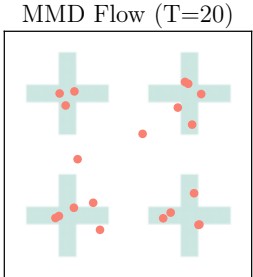 | 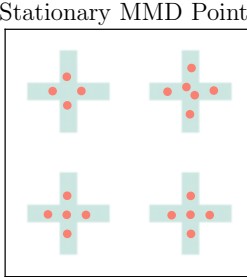 |

**Figure 1.** Approximation of a mixture of four cross-shaped uniform distributions with (**Left**) 20 i.i.d. samples and (**Right**) 20 *stationary MMD points* computed via a discretised gradient flow, simulated for a sufficient length of time $T$ (also shown at $T = 0$ and $T = 20$). *Stationary* MMD points correspond in this case to a local minimum of the MMD, as there are 4 samples in the first cross and 6 samples in the second cross; these points can be explicitly computed. *Minimum* MMD points (not shown) would presumably assign 5 points to each cross, but such points cannot be computed in general.

and are often only guaranteed to find a local minima. This observation hence raises the following question:

*"Why do these points sets exhibit strong numerical integration performance despite only corresponding to local minima of the MMD?"* [1].

**Our Contributions:** We address this question by studying the asymptotic numerical integration error of *stationary MMD points*, defined as point sets[2] $\{\boldsymbol{x}_i\}_{i=1}^n$ for which the associated empirical measure $\mu_n$ satisfies:

**Convergence**: $\mathrm{MMD}(\mu, \mu_n) \to 0$ as $n \to \infty$.

**Stationarity for fixed** $n$: The gradient of $\mathrm{MMD}(\mu, \mu_n)$ with respect to $\{\boldsymbol{x}_i\}_{i=1}^n$ is $\boldsymbol{0}$.

Although minimum MMD points also satisfy **convergence** and **stationarity**, the class of *stationary* MMD points represents a strict relaxation: they can be obtained via a specific noisy gradient descent scheme proposed in Section 4. Our first main result is that stationary MMD points achieve a *faster-than-MMD numerical integration rate* for a wide class of functions $f$ in $\mathcal{H}$ (Theorem 3.4):

$$\left| \tfrac{1}{n} \sum_{i=1}^n f(\boldsymbol{x}_i) - \int f \, \mathrm{d}\mu \right| = o\left(\mathrm{MMD}(\mu, \mu_n)\right). \quad (2)$$

Comparing (2) and (1) shows that numerical integration using stationary MMD points converges faster than the worst-case scenario anticipated by the MMD. This is surprising, but not paradoxical, as the integrand $f$ in (2) is fixed, while $f_n = \frac{1}{n} \sum_{i=1}^n k(\boldsymbol{x}_i, \cdot) - \int k(\boldsymbol{x}, \cdot) \mathrm{d}\mu(\boldsymbol{x})$, the integrand that realises equality in (1), is dependent on the

point set $\{\boldsymbol{x}_i\}_{i=1}^n$. Thus (2) can be interpreted as a novel form of *super-convergence* (see [83] and Remark 3.5). The intuition behind our proof is that stationary MMD points achieve exact integration for a large linear subspace of $\mathcal{H}$, and this subspace expands as $n$ increases.

Our second main result confirms that stationary MMD points can be obtained *exactly* through an interacting particle systems simulated until stationarity (Corollary 4.2). This is in contrast with *minimum* MMD points, which typically cannot be recovered exactly due to the non-convexity of MMD. The particle system considered is derived from MMD gradient flow [3, 54, 38, 30], which formulates an optimisation problem in the space of probability distributions over $\mathbb{R}^d$. To this end, we prove a novel non-asymptotic rate of convergence for the time-discretised, finite-particle MMD gradient flow (Theorem 4.1). Our analysis incorporates the noise injection scheme proposed in [2] and builds upon recent advances in non-asymptotic finite-particle analysis of gradient flows [9], which may be of independent interest. This second main result therefore confirms that MMD gradient flow have potential beyond generative modelling, the task they were originally introduced to tackle. Finally, the effectiveness of stationary MMD points for integration is empirically confirmed.

## 2. Related Work

Before we present our new results, we first review existing work on approximation of probability measures $\mu$ related to MMD. Our focus is restricted to uniformly-weighted empirical measures $\mu_n$; a discussion of alternative methods and results, including those based on weighted empirical measures, is contained in Appendix A. Uniform weights are often preferred due to their simplicity, numerical stability, and robustness to violation of assumptions on the integrand.

**Minimum MMD / Energy Points:** Similar to minimum MMD points, there is a rich literature studying point sets, called *minimum* energy points, that minimise

---

[1]The family of kernel thinning algorithms does not suffer from this gap between theory and practice [40, 39, 72, 86, 24]: these methods enjoy better-than-Monte-Carlo theoretical guarantees which support their empirical performances. However, kernel thinning optimizes over a fixed, finite candidate pool; as a consequence, the quality of the resulting coreset depends on this candidate set, and optimal solutions lying outside the pool cannot be recovered.

[2]To be precise, we consider sequences $(\{\boldsymbol{x}_i^{(n)}\}_{i=1}^n)_{n \in \mathbb{N}}$ of not-necessarily-nested point sets, but to simplify notation we leave the superscript indexing the sequence implicit.

an *energy* functional of the form $\frac{1}{n^2}\sum_{i,j=1}^n \rho(\boldsymbol{x}_i, \boldsymbol{x}_j) - \frac{2}{n}\sum_{i=1}^n \mathbb{E}_{\boldsymbol{y}\sim\mu}[\rho(\boldsymbol{x}_i, \boldsymbol{y})]$. For instance, minimum energy points were studied to construct well-separated nodes over a unit sphere with $\rho(\boldsymbol{x}, \boldsymbol{x}') = \|\boldsymbol{x} - \boldsymbol{x}'\|^{-r}$ for $0 < r < 2$, where $\|\cdot\|$ denotes the Euclidean norm on $\mathbb{R}^d$ [20, 75, 47]. For $\rho(\boldsymbol{x}, \boldsymbol{x}') = -\|\boldsymbol{x} - \boldsymbol{x}'\|$, it was proved in [73] that the integration error of minimum energy points is $\mathcal{O}(n^{-\frac{1}{2}}(\log n)^{-\frac{1}{2d}})$, calling them *support points*. This energy functional coincides (up to an additive constant) with the squared MMD in the special case where $\rho$ is a kernel. As with minimum MMD points, the algorithms proposed in [73, 54] do not come with guarantees of finding minimum energy configurations in general. Our contribution addresses this by analysing the properties of stationary MMD points in Section 3. [88] study the properties of stationary energy points over a unit sphere and prove that stationary energy points are indeed minimum energy points if they are geodesically well-separated.

These minimum-MMD constructions have found applications in deterministic sampling [96, 66], dataset compression and coreset selection [97, 93, 40, 39, 72, 86, 24], numerical integration [28, 31, 47], and, more recently, efficient attention mechanisms [24, 84].

**Quasi Monte Carlo:** Sophisticated quasi Monte Carlo (QMC) point sets have been developed for common distributions $\mu$, typically uniform distributions on a hypercube, $d$-sphere, or simple transformations thereof [36, 64]. Although QMC can exhibit a fast rate of convergence for integration, and these points can be explicitly computed, it does not currently represent a solution for general targets $\mu$ on $\mathbb{R}^d$ (e.g $\mu$ in Figure 1). The traditional discrepancy used in QMC is the star discrepancy, while much of the modern literature adopts the more general perspective of MMD [55].

**Compression:** *Kernel herding* is a sequential algorithm that performs conditional gradient descent on the MMD [26]. It requires solving non-convex optimisation problems for the next point, which can be approximated by searching over a large candidate set $\{\boldsymbol{y}_i\}_{i=1}^N$, but little is known about its theoretical properties except for in finite-dimensional RKHSs [6, 82]. Greedy minimisation of MMD was proposed in [93], again selecting points from a large candidate set $\{\boldsymbol{y}_i\}_{i=1}^N$, with a slower-than-Monte Carlo rate for infinite-dimensional RKHSs. *Kernel thinning* is a compression algorithm that selects $n$ most representative points from a large dataset $\{\boldsymbol{y}_i\}_{i=1}^N$ also by minimizing the MMD [40, 39], achieving faster-than-Monte Carlo rate. The method has been accelerated to near-linear time via *KT-Compress++* [86], extended to adjust for bias in the initial candidate set [72] and further extended to exploit low-rank structures in either the kernel matrix or the data [24]. The main distinction between these algorithms and our stationary MMD points is that we do not constrain our points $\{\boldsymbol{x}_i\}_{i=1}^n$ to be a subset of a candidate

set $\{\boldsymbol{y}_i\}_{i=1}^N$.

**Gradient Flows:** Recently, a line of research on Wasserstein gradient flows frames the approximation of $\mu$ as optimisation of a statistical divergence $\mathcal{D}(\cdot\|\mu)$ over a space of probability distributions. Concerning numerical integration, among the various choices for $\mathcal{D}$ a natural candidate is MMD due to the connection to integration error in (1). The resulting MMD gradient flow was introduced in [3] and has been widely-used in generative modelling [43, 54, 1, 53, 32, 94]. It has also demonstrated strong empirical performance for numerical integration [96, 27, 14]. Unfortunately, a formal theoretical study of the convergence of time-discretised finite-particle MMD gradient flow is missing in the literature—the primary challenge is that MMD is non-convex in the Wasserstein metric [30]. Our second main theoretical contribution addresses this gap in Section 4, which may be of independent interest. Related to this result, [18, 33] recently proved global convergence of MMD gradient flow for the Riesz kernel, under conditions that include boundedness of the logarithms of the candidate and target densities. However, as these results do not apply to the finite particle setting, or to targets supported on non-compact domains, they cannot be used in our context.

In summary, our proposed *stationary MMD points* offers several advantages over existing methods. Compared with quasi–Monte Carlo, it allows greater applicability with fewer constraints on the domain. Compared with the family of kernel thinning algorithms, it remains applicable even when no candidate set is available a priori. Finally, compared with other greedy optimisation–based algorithms, our method comes with strong theoretical guarantees.

## 3. Numerical Integration with Stationary MMD Points

This section establishes our first main theoretical result; super-convergence of stationary MMD points. The analysis of algorithms for computing stationary MMD points is deferred to Section 4.

**Set-Up and Notation:** Let $\mathcal{P}(\mathbb{R}^d)$ denote the set of Borel probability distributions on $\mathbb{R}^d$. Throughout we let $\mu \in \mathcal{P}(\mathbb{R}^d)$ denote the distribution of interest, and $\mathcal{X} = \text{supp}(\mu) \subseteq \mathbb{R}^d$ denote its support. Let $k\colon \mathbb{R}^d \times \mathbb{R}^d \to \mathbb{R}$ be a positive semi-definite kernel and $\mathcal{H}$ be its associated RKHS [4], a Hilbert space of functions on $\mathbb{R}^d$ with inner product $\langle\cdot,\cdot\rangle_{\mathcal{H}}$ and norm $\|f\|_{\mathcal{H}} = \sqrt{\langle f, f\rangle_{\mathcal{H}}}$, containing $\text{span}\{k(\boldsymbol{x}, \cdot) : x \in \mathbb{R}^d\}$ as a dense subset, and for which the *reproducing property* $f(\boldsymbol{x}) = \langle f, k(\boldsymbol{x}, \cdot)\rangle_{\mathcal{H}}$ holds for all $f \in \mathcal{H}$ and $\boldsymbol{x} \in \mathbb{R}^d$.

Given a function $f\colon \mathbb{R}^d \to \mathbb{R}$, let $\partial_\ell f(\boldsymbol{x})$ denote the partial derivative with respect to the $\ell$-th coordinate in $\boldsymbol{x}$

and denote $\nabla f(\boldsymbol{x}) = [\partial_1 f(\boldsymbol{x}), \ldots, \partial_d f(\boldsymbol{x})]^\top \in \mathbb{R}^d$. Let $\partial_\ell k(\boldsymbol{x}, \boldsymbol{x}')$ denote the partial derivative with the $\ell$-th coordinate of $\boldsymbol{x}$ and let $\partial_\ell \partial_{\ell+d} k(\boldsymbol{x}, \boldsymbol{x}')$ denote the mixed partial derivative with respect to the $\ell$-th coordinate in $\boldsymbol{x}$ and the $\ell$-th coordinate in $\boldsymbol{x}'$. Moreover, denote $\nabla_1 k(\boldsymbol{x}, \boldsymbol{x}') = [\partial_1 k(\boldsymbol{x}, \boldsymbol{x}'), \ldots, \partial_d k(\boldsymbol{x}, \boldsymbol{x}')]^\top \in \mathbb{R}^d$.

Throughout the paper, we make the following assumption on the kernel $k$:

**Assumption 1** ($\mu$-integrability and differentiability)**.** The kernel is such that $\sup_{\boldsymbol{x} \in \mathbb{R}^d} \int k(\boldsymbol{x}, \boldsymbol{y}) \, \mathrm{d}\mu(\boldsymbol{y}) < \infty$ and $\sup_{\boldsymbol{x} \in \mathbb{R}^d} \int \partial_\ell k(\boldsymbol{x}, \boldsymbol{y}) \, \mathrm{d}\mu(\boldsymbol{y}) < \infty$ for each $\ell \in \{1, \ldots, d\}$, and $(\boldsymbol{x}, \boldsymbol{y}) \mapsto \partial_\ell \partial_{\ell+d} k(\boldsymbol{x}, \boldsymbol{y})$ is continuous on $\mathbb{R}^d \times \mathbb{R}^d$.

This assumption is satisfied by popular kernels including Gaussian, Laplacian, Matérn, and inverse multiquadric kernels. It is also satisfied by linear kernel $k(\boldsymbol{x}, \boldsymbol{y}) = \boldsymbol{x}^\top \boldsymbol{y}$ when $\mu$ is a mean-zero distribution. The $\mu$-integrability of $k$ ensures that the MMD is well-defined [89]. From Corollary 4.36 of [91], for kernels that satisfy Assumption 1, we know that $\partial_\ell k(\boldsymbol{x}, \cdot) \in \mathcal{H}$ for any $\boldsymbol{x} \in \mathbb{R}^d$ and any $\ell \in \{1, \ldots, d\}$.

The key to our first main result, on super-convergence of the integration error, is an observation that stationary MMD points exactly integrate a linear subspace of $\mathcal{H}$:

$$\mathcal{G}_{\mathcal{X}_n} := \operatorname{span}\{\partial_\ell k(\boldsymbol{x}, \cdot) : \boldsymbol{x} \in \mathcal{X}_n, \ 1 \le \ell \le d\} \subset \mathcal{H},$$
$$\text{where} \quad \mathcal{X}_n = \{\boldsymbol{x}_i\}_{i=1}^n \subset \mathbb{R}^d. \tag{3}$$

as shown in the following proposition:

**Proposition 3.1** (Exactness of numerical integration)**.** *Suppose the kernel satisfies Assumption 1. Then integration using $\{\boldsymbol{x}_i\}_{i=1}^n$ is exact on $\mathcal{F}_n := \operatorname{span}(\{1\} \cup \mathcal{G}_{\mathcal{X}_n})$. That is, $\frac{1}{n}\sum_{i=1}^n f(\boldsymbol{x}_i) = \int f \mathrm{d}\mu$ for all $f \in \mathcal{F}_n$.*

*Proof of Proposition 3.1* Since the weights sum to 1, $\mu_n$ is exact on the constant functions, $\operatorname{span}(\{1\})$. We are thus left to check the exactness on $\mathcal{G}_{\mathcal{X}_n}$. The **stationarity** condition reads, for each $i \in \{1, \ldots, n\}$,

$$\nabla_{\boldsymbol{x}_i} \operatorname{MMD}^2\left(\mu, \frac{1}{n}\sum_{j=1}^n \delta_{\boldsymbol{x}_j}\right) = \boldsymbol{0} \iff$$
$$\frac{1}{n}\sum_{j=1}^n \nabla_1 k(\boldsymbol{x}_i, \boldsymbol{x}_j) - \int \nabla_1 k(\boldsymbol{x}_i, \boldsymbol{y}) \mathrm{d}\mu(\boldsymbol{y}) = \boldsymbol{0}, \tag{4}$$

where we have interchanged $\mu$-integral and derivative (justified under Assumption 1; see Lemma B.1). This final equation shows that $\mu_n$ is exact on $\boldsymbol{x} \mapsto \partial_\ell k(\boldsymbol{x}_i, \boldsymbol{x})$ for each $\ell \in \{1, \ldots, d\}$ and $i \in \{1, \ldots, n\}$, completing the argument. $\qquad\square$

**Remark 3.2** (Example of $\mathcal{G}_{\mathcal{X}_n}$)**.** To gain more insight into the space $\mathcal{G}_{\mathcal{X}_n}$, consider a polynomial kernel $k(\boldsymbol{x}, \boldsymbol{y}) = (\boldsymbol{x}^\top \boldsymbol{y} + 1)^r$, where $r \in \mathbb{N}$. In this setting, let $\varphi_r : \mathbb{R}^d \to \mathbb{R}^m$ denote the feature map consisting of all monomials of total degree $\le r$ (excluding constant functions), where $m = \binom{d+r}{r} - 1$. If the matrix $\Phi_r := [\varphi_r(\boldsymbol{x}_1), \ldots, \varphi_r(\boldsymbol{x}_n)] \in$

$\mathbb{R}^{m \times n}$ has rank $m$, then $\mathcal{G}_{\mathcal{X}_n}$ coincides with the space of all polynomial functions of degree $\le r$ (excluding constant functions), meaning the numerical integration estimator is *polynomially exact*. This can be contrasted with [61], which relies on non-uniform weights to minimise MMD (for a fixed point set) subject to a polynomial exactness constraint.

Given an integrand $f$, we can consider a decomposition $f = f_n + (f - f_n)$ where $f_n \in \mathcal{F}_n$ is exactly integrated by Proposition 3.1. The integration error therefore depends only on the difficulty of numerically integrating the remainder term $f - f_n$. This observation motivates us to investigate the approximation capacity of $\mathcal{F}_n$ with respect to the whole RKHS $\mathcal{H}$. To this end, we make the following assumptions:

**Assumption 2** (Connected support)**.** For any $\boldsymbol{x}, \boldsymbol{y} \in \mathcal{X}$, there exists a piecewise continuously differentiable curve $\gamma : [0, 1] \to \mathcal{X}$ with $\gamma(0) = \boldsymbol{x}$ and $\gamma(1) = \boldsymbol{y}$.

This assumption holds if $\mathcal{X}$ is the closure of an open connected set or an embedded $C^1$ manifold, which includes common sets such as $\mathbb{R}^d$, spheres or spherical surfaces as examples. This assumption can be relaxed to $\mathcal{X}$ being the finite union of sets satisfying Assumption 2 (as illustrated in Figure 1) if $\mathcal{H}$ does not contain non-zero constant functions on the subsets, as is the case for Gaussian RKHSs and subsets with non-empty interiors [91, Corollary 4.44].

**Assumption 3** ($C_0$-universality)**.** The kernel $k$ is $C_0$-universal; i.e., the associated RKHS $\mathcal{H}$ has a subset that is dense in $C_0(\mathbb{R}^d)$, the space of continuous functions $f : \mathbb{R}^d \to \mathbb{R}$ that vanish at infinity, with respect to the norm $\|f\|_\infty = \sup_{\boldsymbol{x} \in \mathbb{R}^d} |f(\boldsymbol{x})|$.

The reader is referred to [23, 90] for a detailed discussion of $C_0$-universal kernels. Assumption 1 and 3 are satisfied by Gaussian, Laplacian, Matérn, and inverse multiquadric kernels.

Proposition 3.3 below formalizes the approximation capacity of $\mathcal{F}_n$. To state this result, we let $\mathcal{H}_{\mathcal{X}}$ denote the closure of $\operatorname{span}\{k(\boldsymbol{x}, \cdot) : \boldsymbol{x} \in \mathcal{X}\}$ in $\mathcal{H}$.

**Proposition 3.3** (Asymptotic approximation capacity of $\mathcal{F}_n$)**.** *Suppose $\mathcal{X}$ satisfies Assumption 2 and $k$ satisfies Assumption 1 and 3. Let $f = c + h$ with $c \in \mathbb{R}$, $h \in \mathcal{H}_{\mathcal{X}}$, and consider the semi-norm*

$$|f|_n := \inf\{\|r_n\|_{\mathcal{H}} : f = f_n + r_n, f_n \in \mathcal{F}_n, r_n \in \mathcal{H}\} \tag{5}$$

*If $h$ is non-constant on $\mathcal{X}$, then $\lim_{n \to \infty} |f|_n = 0$.*

*Proof of Proposition 3.3.* For any $\varepsilon > 0$, we show $|f|_n < \varepsilon$ for sufficiently large $n \in \mathbb{N}$. The key observation is that we can approximate $h$ using an element of $\mathcal{G}_{\mathcal{X}} := \operatorname{span}\{\partial_\ell k(\boldsymbol{x}, \cdot) : \boldsymbol{x} \in \mathcal{X}, 1 \le \ell \le d\}$, which can be further approximated by an element of $\mathcal{G}_{\mathcal{X}_n}$. The denseness of $\mathcal{G}_{\mathcal{X}_n}$ follows from the denseness of stationary MMD points

in $\mathcal{X}$, due to its **convergence** property. These claims are proved in Lemma B.5 and Lemma B.6 (see Appendices B.2 and B.3). Formally, by Lemma B.5, there exists $g_\varepsilon \in \mathcal{G}_\mathcal{X}$ satisfying $\|h - g_\varepsilon\|_\mathcal{H} < \varepsilon/2$. Also, Lemma B.6 states that for sufficiently large $n \in \mathbb{N}$, we may take $g_{n,\varepsilon} \in \mathcal{G}_{\mathcal{X}_n}$ such that $\|g_\varepsilon - g_{n,\varepsilon}\|_\mathcal{H} < \varepsilon/2$. Since $f = c + g_{n,\varepsilon} + h - g_\varepsilon + g_\varepsilon - g_{n,\varepsilon}$, the definition of $|f|_n$ in (5) implies $|f|_n \le \|h - g_\varepsilon\|_\mathcal{H} + \|g_\varepsilon - g_{n,\varepsilon}\|_\mathcal{H} < \varepsilon$. $\qquad\square$

The subset $\mathcal{H}_\mathcal{X} \subset \mathcal{H}$ consists of functions that do not vanish on $\mathcal{X}$ (see Remark B.3). Proposition 3.3 therefore states that $\mathcal{G}_{\mathcal{X}_n}$ can approximate functions in $\mathcal{H}$ with non-trivial $\mu$-integrals increasingly well as the sample size $n$ is increased, as measured in the seminorm $|\cdot|_n$. We can now state our main theorem, which shows that the integration error of stationary MMD points vanishes faster than MMD:

**Theorem 3.4** (Super-convergence of stationary MMD points). *Suppose the kernel satisfies Assumption 1. Then, $\forall n \in \mathbb{N}$, the integration error of a stationary MMD point set $\{\boldsymbol{x}_i\}_{i=1}^n$ for an integrand $f \in \mathrm{span}(\{1\} \cup \mathcal{H})$ satisfies*

$$\left| \tfrac{1}{n} \sum_{i=1}^n f(\boldsymbol{x}_i) - \int f \,\mathrm{d}\mu \right| \le |f|_n \cdot \mathrm{MMD}\,(\mu, \mu_n). \quad (6)$$

*where $\mu_n = n^{-1} \sum_{i=1}^n \delta_{\boldsymbol{x}_i}$. Furthermore, suppose $\mathcal{X}$ satisfies Assumption 2 and $k$ satisfies Assumption 3. Let $f = c + h$ with $c \in \mathbb{R}$ and $h \in \mathcal{H}_\mathcal{X}$ non-constant on $\mathcal{X}$. Then $|f|_n \to 0$ as $n \to \infty$, so the numerical integration estimator achieves a super MMD error rate $o\,(\mathrm{MMD}(\mu, \mu_n))$.*

*Proof of Theorem 3.4.* The proof exploits the control variate trick of [8]. The integrand $f$ can be written as $f = f_n + r_n$ with $f_n \in \mathcal{F}_n$, whence

$$\left| \tfrac{1}{n} \sum_{i=1}^n f(\boldsymbol{x}_i) - \int f \,\mathrm{d}\mu \right| = \left| \tfrac{1}{n} \sum_{i=1}^n r_n(\boldsymbol{x}_i) - \int r_n \,\mathrm{d}\mu \right|$$
$$\le \|r_n\|_\mathcal{H} \cdot \mathrm{MMD}\,(\mu, \mu_n),$$

where the equality follows since $f_n \in \mathcal{F}_n$ is exactly integrated (see Proposition 3.1) and the inequality is Cauchy–Schwarz. Taking the infimum over all possible choices of $r_n$ in the decomposition of $f$ yields $|\tfrac{1}{n} \sum_{i=1}^n f(\boldsymbol{x}_i) - \int f \,\mathrm{d}\mu| \le |f|_n \cdot \mathrm{MMD}(\mu, \mu_n)$. Proposition 3.3 proves that $\lim_{n\to\infty} |f|_n = 0$ when $f = c + h$ with $c \in \mathbb{R}$ and $h \in \mathcal{H}_\mathcal{X}$ non-constant on $\mathcal{X}$, completing the argument. $\qquad\square$

**Remark 3.5** (Super-convergence). In approximation theory, the term *super-convergence* usually refers to how a worst-case error bound, such as (1), can be bypassed if additional regularity can be exploited. A result analogous to Theorem 3.4 can be trivially deduced for optimal *non-uniform* weights (see Appendix A and [42, 59]); one interpretation of Theorem 3.4 is therefore that non-uniform weights are not required to attain super-convergence for numerical integration.

**Remark 3.6** (Non-uniqueness). Stationary MMD points are not necessarily unique, even up to permutation of the particle labels; however, this non-uniqueness is not problematic, since our integration guarantees apply to *any* stationary configuration reached by the particle system.

## 4. Computing Stationary MMD Points

This section contains our second main contribution: a practical implementation of stationary MMD points through an MMD gradient flow. MMD gradient flows start by initialising particles $\{\boldsymbol{x}_i^{(1)}\}_{i=1}^n \subset \mathbb{R}^d$ and then, for kernels that satisfy Assumption 1, these particles are updated via gradient descent on the (squared) MMD: for any $t \in \{1, \dots, T\}$,

$$\boldsymbol{x}_i^{(t+1)} = \boldsymbol{x}_i^{(t)} - \gamma \Big\{ \tfrac{1}{n} \sum_{j=1}^n \nabla_1 k(\boldsymbol{x}_i^{(t)}, \boldsymbol{x}_j^{(t)})$$
$$- \mathbb{E}_{\boldsymbol{Y} \sim \mu}[\nabla_1 k(\boldsymbol{x}_i^{(t)}, \boldsymbol{Y})] \Big\} \quad (7)$$

where $\gamma > 0$ is a step size at time $t$ to be specified. The update (7) can equivalently be viewed as a discretised Wasserstein gradient flow on the MMD [3, Equation 21], and thus it decreases MMD in the steepest direction in terms of both the Euclidean and Wasserstein geometry: $\mathrm{MMD}(\mu, \hat{\nu}_n^{(t+1)}) \le \mathrm{MMD}(\mu, \hat{\nu}_n^{(t)})$ where $\hat{\nu}_n^{(t)}$ is the empirical measure of the particles at time $t$, provided the step size $\gamma$ is small enough (see Proposition 4 of [3] and our Lemma B.7). Naturally, if we simulate (7) for long enough the **stationarity** property will be satisfied; this is verified numerically in Section 5.

Unfortunately, even though (7) is guaranteed to decrease MMD at each step, the MMD gradient flow $(\hat{\nu}_n^{(t)})_{i=1}^\infty$ might still *fail to converge* to the target distribution $\mu$ even with infinite time steps and infinite number of particles, which violates the first **convergence** property of the stationary MMD points. Indeed, this is a well-known challenge for MMD gradient flows (7) due to lack of convexity of MMD with respect to the Wasserstein-2 metric [30]. As a result, there exists a gap between theoretical results and practical algorithms that minimise the MMD [96, 73, 27].

To improve convergence of MMD gradient flow, we follow [3] and inject noise into (7). Specifically, with $(\boldsymbol{u}_i^{(t)})_{i=1}^n \sim \mathcal{N}(0, \mathrm{I}_d)$ and $\beta_t > 0$ for $t \in \mathbb{N}$,

$$\boldsymbol{x}_i^{(t+1)} = \boldsymbol{x}_i^{(t)} - \gamma \Big\{ \tfrac{1}{n} \sum_{j=1}^n \nabla_1 k(\boldsymbol{x}_i^{(t)} + \beta_t \boldsymbol{u}_i^{(t)}, \boldsymbol{x}_j^{(t)})$$
$$- \mathbb{E}_{\boldsymbol{Y} \sim \mu}[\nabla_1 k(\boldsymbol{x}_i^{(t)} + \beta_t \boldsymbol{u}_i^{(t)}, \boldsymbol{Y})] \Big\}. \quad (8)$$

We call the above update scheme (8) *noisy MMD particle descent* with noise scale $\beta_t$, and we will show below that it indeed converges to *stationary* MMD points as desired. It is a key distinction relative to minimum MMD points, which cannot be computed in general.

To implement (8) the expectation $\mathbb{E}_{Y \sim \mu}[\nabla_1 k(\cdot, Y)]$ is required. For certain kernels $k$ and targets $\mu$ this can be exactly computed; see, for example, [21] for a comprehensive list of closed-form kernel mean embeddings which can then be differentiated, or [31] for the use of change-of-variable techniques to facilitate the computation. When $\mu$ is only known up to normalisation, Stein reproducing kernels provide a viable alternative [77]. Moreover, when $\mu$ corresponds to an empirical distribution supported on a dataset, $\mathbb{E}_{Y \sim \mu}[\nabla_1 k(\cdot, Y)]$ can be computed trivially, since the expectation reduces to an empirical average over the dataset. If this expectation is instead approximated by an unbiased estimator with controlled variance, for instance through subsampling, then an additional variance term would appear in the upper bound of Theorem 4.1.

A novel convergence analysis of noisy MMD particle descent (7) is our second main contribution. In particular, we show that (7) indeed finds the *stationary* MMD points under the following assumptions:

**Assumption 4** (Kernel regularity). The maps $x \mapsto k(x, x)$, $x \mapsto \partial_\ell \partial_{\ell+d} k(x, x)$, and $x \mapsto \partial_\ell \partial_r \partial_{r+d} \partial_{\ell+d} k(x, x)$ are continuous and bounded, uniformly over $r, \ell \in \{1, \ldots, d\}$, by a constant $\kappa$.

Assumption 4 is a standard assumption for Wasserstein gradient flows with smooth kernels [46, 3, 52, 65]. Many kernels satisfy Assumption 1, 3 and 4, including the Gaussian kernel, Matérn kernel of order $\zeta$ with $\zeta + d/2 \geq 2$, and the inverse multiquadric kernel.

**Assumption 5** (Noise injection level). Define $\Phi(z, w) = \mathbb{E}_{Y \sim \mu}[\nabla_1 k(z, Y)] - \nabla_1 k(z, w)$. For each $t \in \mathbb{N}^+$, the noise injection level $\beta_t \in (0, 1)$ satisfies, with $\underline{u}^{(t)} := (u_i^{(t)})_{i=1}^n \overset{\text{i.i.d.}}{\sim} \mathcal{N}(0, \mathrm{I}_d)$,

$$\frac{1}{n} \sum_{i=1}^n \mathbb{E}_{\underline{u}^{(t)}} \left[ \left\| \frac{1}{n} \sum_{j=1}^n \Phi(x_i^{(t)} + \beta_t u_i^{(t)}, x_j^{(t)}) \right\|^2 \right]$$
$$\geq 4\beta_t^2 d^2 \kappa^2 \, \mathrm{MMD}^2 \left( \frac{1}{n} \sum_{i=1}^n \delta_{x_i^{(t)}}, \mu \right). \tag{9}$$

Assumption 5 states that with appropriate control of noise injection level $\beta_t$, the noisy MMD particle descent satisfies a *gradient dominance* condition in expectation (also known as the Polyak–Lojasiewicz inequality), a weaker condition than convexity to ensure fast global convergence [19]. *A posteriori* verification of Assumption 5 is possible by drawing many realisations of the injected noise and estimate the left-hand side in (9) with an empirical average. Since $\Phi$ is bounded from Assumption 4, this empirical estimate is expected to concentrate fast to its expectation. In practice, at each iteration $t-1$, one can ensure that Assumption 5 is satisfied by selecting $\beta_t$ from a pre-specified candidate set

(e.g., $\{t^{-0.5}, t^{-0.25}, t^{-0.1}\}$) based on an empirical check of which choice meets the condition. This extra selection step, however, increases the per-iteration computational cost.

Similar assumptions to Assumption 5 have been widely used in the convergence analysis of kernel-based gradient flows and other related non-convex optimisation; see Proposition 8 of [3], Proposition 5 of [46] and [51]. However, their assumptions on $\beta_t$ involve expectations with respect to $\{x_i^{(t)}\}_{i=1}^n$ on both sides of (9), which means they need to be checked for *any* possible trajectory. In contrast, our assumption only needs to be checked for the current realized trajectory. Denoting $a \vee b = \max\{a, b\}$, we have:

**Theorem 4.1.** *Suppose the kernel $k$ satisfies Assumption 1 and 4. Given initial particles $\{x_i^{(1)}\}_{i=1}^n$, suppose the noisy MMD particle descent defined in (8) satisfies Assumption 5 and the step size $\gamma$ satisfies $256\gamma^2 d^2 \kappa^2 \leq 1$ for any $t \in \mathbb{N}^+$. Denote $\mathcal{S}(t) = \mathrm{MMD}(\frac{1}{n} \sum_{i=1}^n \delta_{x_i^{(t)}}, \mu)$ for any $t \in \mathbb{N}^+$.*

*Then for any $T \in \mathbb{N}^+$, and for any $\delta > 0$, we have with probability at least $1 - \exp(-\delta)$ that*

$$\mathcal{S}(T) \leq \exp\left( -\frac{\kappa d^2}{2} \sum_{t=1}^{T-1} \gamma \beta_t^2 \right) \mathcal{S}(1) + \frac{\delta + \log T}{\sqrt{n}}. \tag{10}$$
$$\left( C \sum_{t=1}^{T-1} \left( (\gamma \beta_t)^{\frac{1}{2}} \vee n^{-\frac{1}{2}} \right) \cdot \exp\left( -\frac{\kappa d^2}{2} \sum_{s=t+1}^{T-1} \gamma \beta_s^2 \right) \right)$$

*where $C > 0$ is a universal constant independent of $n, (\beta_t)_{t=1}^T, T$.*

The proof of Theorem 4.1 can be found in Appendix B.4. The right-hand side of (10) consists of two terms: an optimisation error and a finite-sample estimation error. The optimisation error captures the decay of the MMD from its initial value under the dynamics of noisy MMD particle descent. Its exponential decay rate is a direct consequence of the gradient dominance condition from Assumption 5, and this term vanishes when $\sum_{t=1}^{T-1} \beta_t^2 \to \infty$ as $T \to \infty$, recovering the convergence result in Proposition 8 of [3]. The latter estimation error quantifies the statistical error incurred from using $n$ particles. The $1/\sqrt{n}$ scaling factor arises from a standard concentration inequality, as established in Lemma B.8. We highlight a trade-off between the optimisation and estimation errors: increasing the noise level $\beta_t$ accelerates the convergence (reducing optimisation error) but simultaneously increases the variance of the updates, leading to larger estimation error.

The exact non-asymptotic rate of convergence of the noisy MMD particle descent largely depends on the noise injection scheme $\beta_t$. Corollary 4.2, which is proved in Appendix B.4, provides further insight:

**Corollary 4.2.** *In the setting of Theorem 4.1, suppose that there exists $\beta_t \propto t^{-\alpha}$ for $0 \leq \alpha < 1/2$ such that Assumption 5 holds. Define the constant $C' = 0.5\gamma \kappa^2 d^2$. Then, for*

*any* $T \leq n^{\frac{1}{\alpha}}$,

$$\mathrm{MMD}(\hat{\nu}_n^{(T)}, \mu) = \mathcal{O}_P\left(\exp\left(-C'T^{1-2\alpha}\right) + \frac{(\log T)^2 T^{\alpha}}{\sqrt{n}}\right)$$

*where* $\hat{\nu}_n^{(T)} = \frac{1}{n}\sum_{i=1}^{n}\delta_{\boldsymbol{x}_i^{(T)}}$.

By choosing $T = \mathcal{O}((\log n)^{\frac{1}{1-2\alpha}})$, we obtain the convergence rate $\mathrm{MMD}(\hat{\nu}_n^{(T)}, \mu) = \mathcal{O}_P(n^{-\frac{1}{2}})$ up to some logarithm factors, which implies that the particles $\{\boldsymbol{x}_i^{(T)}\}_{i=1}^{n}$ satisfy the **convergence** property in the $n, T \to \infty$ limit. Furthermore, since $\lim_{t\to\infty}\beta_t = 0$, the particles $\{\boldsymbol{x}_i^{(T)}\}_{i=1}^{n}$ obtained from the noisy MMD particle descent (8) also satisfy the **stationarity** property in the same limit. Therefore, Corollary 4.2 establishes that noisy MMD particle descent in (8) can *indeed compute stationary MMD points*, as desired.

The computational cost of stationary MMD points simulated via noisy MMD particle descent is $\mathcal{O}(n^2 T)$. Following the argument above, it suffices to take $T = \mathcal{O}((\log n)^{\frac{1}{1-2\alpha}})$ so the overall computational cost is $\mathcal{O}(n^2)$ up to logarithm factors. Compared to quasi Monte Carlo (QMC), which is computationally cheaper, stationary MMD points offer greater generality, as they can be applied to any target distribution $\mu$ supported on any domain $\mathcal{X}$ that satisfies Assumption 2. Compared to existing approaches based on minimising the MMD or energy functionals, which share the same computational complexity as our stationary MMD points and often claim fast convergence rates [73, 96], we highlight a gap between these theoretical guarantees and the practical issue of their proposed algorithms being trapped in local minima or stationary points. Our *stationary* MMD points are designed to bridge this gap. One can deduce $o_P(n^{-1/2})$ convergence rates for the numerical integration error by combining the MMD convergence rate from Corollary 4.2 and the super-convergence result proved in Theorem 3.4.

**Remark 4.3** (Comparison with existing convergence results in [3]). Our Theorem 4.1 provides the *first* convergence result for time-discretised finite-particle MMD gradient flow with smooth kernels. In contrast, Proposition 8 of [3] proves convergence for the time-discretized population MMD flow, while Theorem 9 of [3] proves convergence of the time-discretized finite-particle MMD flow to its population limit, under the condition $T \geq n\gamma$ fixed a fixed step size $\gamma$. However, gluing these existing results together gives $\mathrm{MMD}(\hat{\nu}_n^{(T)}, \mu) = \mathcal{O}(\exp(-C'T^{1-2\alpha}) + n^{-0.5}\exp(C''T))$ which does not converge when $T \geq n\gamma$.

**Remark 4.4** (Finite particle error bound). Establishing convergence for finite-particle implementations of Wasserstein gradient flows is a challenging task: usually the finite-particle upper-bound grows exponentially with $T$; see for example [3, Theorem 9], [87, Theorem 3], and [30, Proposition 10.1]. In contrast, in Corollary 4.2, our upper-bound is polynomial and even logarithmic in $T$ when $\alpha = 0$. The key insight of our Theorem 4.1 and Corollary 4.2, borrowed

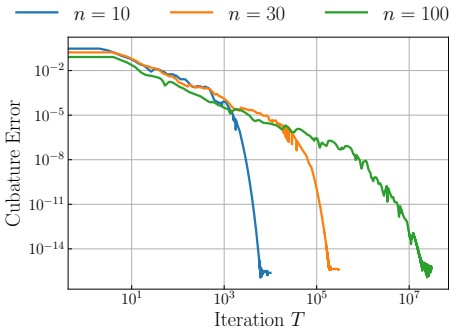

**Figure 2.** Exact integration of a function in $\mathcal{F}_n$ with stationary MMD points computed by MMD gradient flow, verifying Proposition 3.1. The convergence plateaus at around $10^{-15}$ due to numerical precision.

from [9, 92], is to work directly with the joint distribution of the particles and track the evolution of the squared MMD.

**Remark 4.5** ($\tilde{o}(n^{-1})$ integration error with good initialization). Suppose we are given initial points $\{\boldsymbol{x}_i^{(1)}\}_{i=1}^{n}$ with $\mathcal{S}(1) = \mathcal{O}(n^{-1}(\log n)^b)$ for some constant $b$ (e.g via kernel thinning [40, 86]), where $b$ depends on the tail of the distribution $\mu$. Running the finite-particle MMD descent scheme with $\gamma \cdot \beta_t = n^{-1}$ for $T = \mathcal{O}((\log n)^b)$ iterations yields the MMD convergence rate $\mathrm{MMD}(\hat{\nu}_n^{(T)}, \mu) = \mathcal{O}(n^{-1}(\log n)^b)$ (Corollary B.10). Therefore, from Theorem 3.4, the single function integration rate is $o(n^{-1}(\log n)^b)$. We posit the little-o improvement occurs in the log term $(\log n)^b$ rather than in $n^{-1}$. The role of the MMD gradient flow is thus a perturbation on the initial point set by enforcing a **stationarity** property, thereby enhancing numerical integration performance.

## 5. Experiments

This section numerically studies the numerical integration properties of stationary MMD points computed via the noisy MMD particle descent scheme in (8). The code can be found at https://github.com/hudsonchen/stationary_mmd.

### 5.1. Mixture of Gaussians

First we consider a synthetic experiment with the target distribution $\mu = \frac{1}{10}\sum_{m=1}^{10}\mathcal{N}(\boldsymbol{\mu}_m, \boldsymbol{\Sigma}_m)$, a mixture of 10 two-dimensional ($d = 2$) Gaussian distributions following the set-up in [26, 57]. This simple multimodal setting evaluates the ability of sampling methods to capture all modes of a distribution—something i.i.d. samples often fail to achieve, as they tend to assign an imbalanced number of samples across the mixture components. Our stationary MMD points are simulated with noisy MMD particle descent in (8) until stationarity is reached. We use a fixed step size $\gamma = 1.0$ and noise injection level $\beta_t = t^{-1/2}$ such

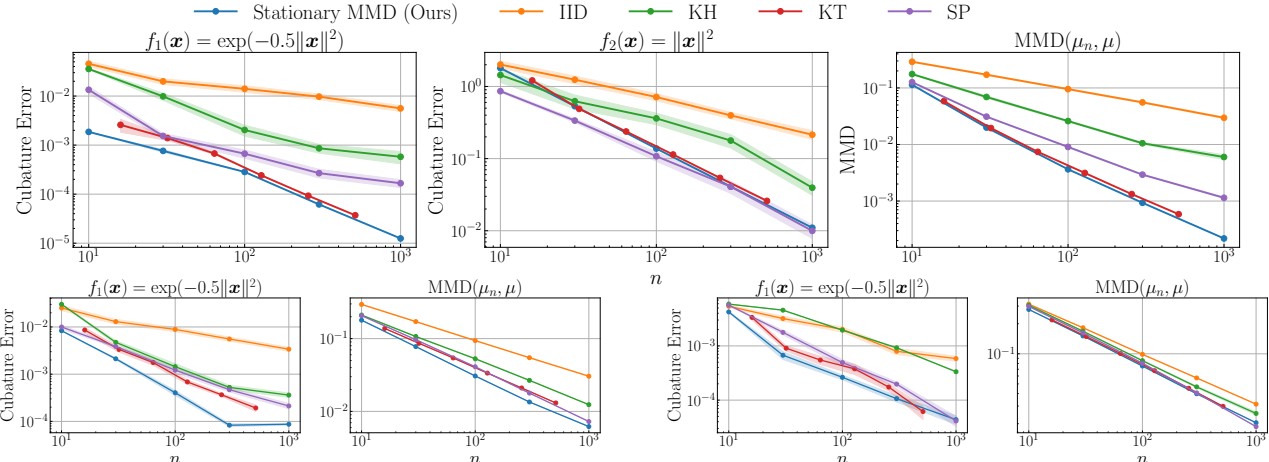

**Figure 3.** Comparison of stationary MMD points with baseline methods. **Top row:** mixture of Gaussians. **Bottom left:** *House8L* dataset. **Bottom right:** *Elevators* dataset. All results are averaged over 20 independent runs with different random seeds; shaded regions indicate 25%–75% quantiles.

that Assumption 5 is empirically satisfied (see Figure 4). All particles were initialised at $\mathbf{0}$ and a Gaussian kernel $k(\boldsymbol{x}, \boldsymbol{y}) = \exp(-0.5\|\boldsymbol{x} - \boldsymbol{y}\|^2)$ was used. The Gaussian kernel satisfies all Assumption 1, 3 and 4 required for the theoretical rate of stationary MMD points to hold.

**Stationarity:** We verify that stationary MMD points indeed satisfy the exact integration property for integrands in $\mathcal{F}_n$. To this end, we consider a function $f(\boldsymbol{x}) = \sum_{j=1}^{n} \sum_{\ell=1}^{d} \partial_\ell k(\boldsymbol{x}_j, \boldsymbol{x})$. This integrand has a $\mu$-integral that can be computed analytically; see Appendix C for the derivations. We see in Figure 2 that integration error indeed vanishes when the noisy particle MMD descent scheme in (8) is simulated until stationarity. This empirically confirms the claim in Proposition 3.1. Although the noisy MMD particle descent scheme requires many iterations to reach convergence, we observe—consistent with Corollary 4.2—that it takes fewer iterations to have good integration performance, as we demonstrate in the following results.

**Integration Benchmark:** Next, we consider numerical integration of the functions $f_1(\boldsymbol{x}) = \exp(-0.5\|\boldsymbol{x}\|^2)$ and $f_2(\boldsymbol{x}) = \|\boldsymbol{x}\|^2$ against the distribution $\mu$. In this simple setting we have closed form expression for both integrals, which allows benchmarking against an array of baseline methods: IID represents identical independent samples, QMC represents quasi Monte Carlo samples, KT represents kernel thinning [40], KH represents kernel herding [26], and SP represents support points [73]. The baselines KT here use *centered* Gaussian kernels and oversampling parameters $\mathfrak{g} = 6$. More implementation details of all baselines can be found in Appendix C. Additional experiments comparing KT between centered and uncentered kernels are presented in Appendix C.2. We also report $\mathrm{MMD}(\mu_n, \mu)$ which is the worse case integration error for all functions in the RKHS $\mathcal{H}$.

In **Top left** and **Top middle** of Figure 3, we observe that stationary MMD points consistently outperform all baselines in terms of integration error for $f_1$, while achieving comparable performance to support points—and outperforming the remaining baselines—for $f_2$. This behavior is expected: $f_1 \in \mathcal{H}$ so stationary MMD points are particularly well-suited for accurate integration; in contrast, $f_2 \notin \mathcal{H}$ which explains the drop in relative performance. It is worth noting that, while KT can achieve performance comparable to our stationary MMD points after careful hyperparameter tuning (with $\mathfrak{g} = 6$), it is substantially more computationally expensive. This overhead arises from the need to select a representative subset of size $n$ from a much larger candidate pool of size $2^{\mathfrak{g}} n^2$. In particular, for $n = 100$, computing stationary MMD points takes approximately 20 seconds, whereas KT requires more than 3000 seconds in this setting. The strong performance of support points for $f_2$ can be attributed to the fact that support points minimise the energy distance with $\rho(\boldsymbol{x}, \boldsymbol{y}) = -\|\boldsymbol{x} - \boldsymbol{y}\|$ [85], which is analogous to MMD with an unbounded kernel, and hence support points may be more suitable for an unbounded integrand. The empirical convergence rate can be estimated by linear regression in log-log space. The integration error with $f_1 \in \mathcal{H}$ exhibits a rate of $-1.39$, while the MMD error decays at a rate of $-1.35$, which confirms the super-convergence predicted by Theorem 3.4.

Finally, to further assess the performance of stationary MMD points in higher dimensions, we conduct experiments on Gaussian mixture models with $d = 10$ and $d = 50$. The results, summarized in Table 1, demonstrate that our method consistently outperforms i.i.d. sampling across all sample sizes and dimensions, which verifies the potential effectiveness of stationary MMD point sets in high-dimensional

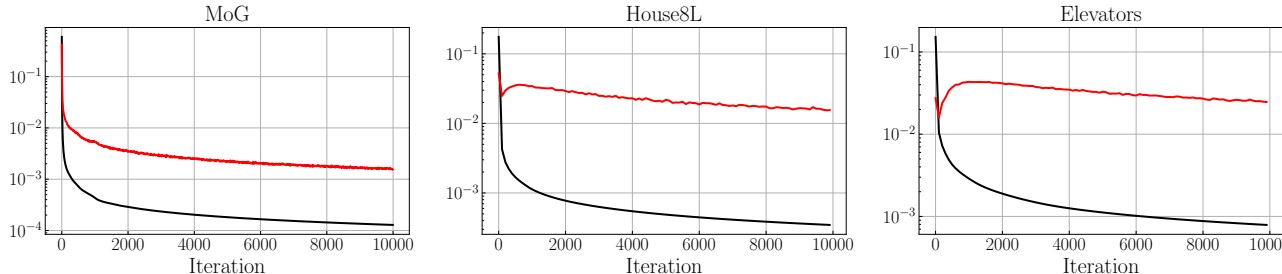

**Figure 4.** Empirical verification of Assumption 5 on noise injection level $\beta_t$ used in all our experiments. We take $n = 100$. The red line represents the left hand side of (9) and the black line represents the right hand side of (9). The figure confirms that the assumption is satisfied for most iterations of the algorithm, with the exception of the first few iterations.

settings. [3]

### 5.2. OpenML Datasets

For our final experiment we considered the *House8L* ($d = 8$) and *Elevators* ($d = 18$) data from OpenML [16], where the target $\mu$ is the empirical measure of the dataset. In this context, approximating $\mu$ by a discrete measure is commonly referred to as coreset selection [7]. Unlike standard coreset methods, our stationary MMD point set—obtained by noisy MMD particle descent simulated for $T = 10,000$ iterations—does not consist of a subset of the original dataset. Although the support of $\mu$ does not meet Assumption 2, our proposed algorithm can still be applied, and it is of interest to evaluate its empirical performance in such settings. We used a Gaussian kernel $k(\boldsymbol{x}, \boldsymbol{y}) = \exp(-0.5\|\boldsymbol{x} - \boldsymbol{y}\|^2)$ and a fixed step size, $\gamma = 1.0$ for *House8L* and $\gamma = 0.3$ for *Elevators*. The noise injection level $\beta_t = t^{-1/2}$ empirically satisfied Assumption 5 (see Figure 4 in Appendix C). All particles were initialised at $\mathbf{0}$, reflecting the fact that the datasets are normalised. In **Bottom** of Figure 3 we observe that stationary MMD points consistently outperform all baselines in terms of integration error for $f_1$ and MMD. The closest baseline being KT which is computationally expensive. Moreover, since the total number of available samples is fixed, constructing a candidate pool of size $2^{\mathfrak{g}} n^2$ requires sampling with replacement. We do not include $f_2$ here as $f_2 \notin \mathcal{H}$. The improvement of stationary MMD points and all baselines are greater in *House8L* than *Elevators*, which we attribute to the lower dimensionality. To investigate the robustness of these results, Figure 5 in Appendix C presents an ablation study comparing the performance of integration methods based instead on the Matérn-$\frac{3}{2}$ kernel. For stationary MMD points, we observed that the Matérn-$\frac{3}{2}$ outperforms the Gaussian kernel on the *House8L* dataset. Practical tools exist for kernel choice, but analyzing their performance for stationary MMD points is beyond our scope.

---

[3]Compared with the empirical results of the first version of this paper [29], the baselines KT achieve much better performances in Figure 3 as a consequence of using *centered* kernels and oversampling hyperparameter $\mathfrak{g} = 6$. See Appendix C.2 for details.

## 6. Discussion

This paper resolves an important open issue in quantisation of distributions and related numerical integration routines; how to reconcile the theoretical performance of minimum MMD points with the reality that only *stationary* MMD points can be computed. Our analysis revealed the surprising result that the integration error using stationary MMD points vanishes *faster* than the MMD – so-called *superconvergence* – providing for the first time an explanation of the strong empirical performance that had been previously observed. In addition, we have substantially strengthened the convergence analysis for noisy MMD particle descent, proving that stationary MMD points can be computed. Recent work has demonstrated the practical utility of MMD-based coreset constructions in large-scale deep learning systems [24], and the stationary MMD point sets proposed here are directly applicable in such contexts—a promising avenue for future work.

## Acknowledgments

ZC was supported by the Engineering and Physical Sciences Research Council (EPSRC) EP/S021566/1. HK and CJO were supported by EP/W019590/1. CJO was supported by a Philip Leverhulme Prize PLP-2023-004. FXB was supported by EPSRC EP/Y022300/1. TK was supported by the Research Council of Finland grant 359183 (Flagship of Advanced Mathematics for Sensing, Imaging and Modelling). The authors would like to thank Matthew Fisher for his helpful comment on the support assumption. FXB would like to thank Mingtian Zhang for encouraging ZC to take up a PhD at UCL. The authors are grateful to Lester Mackey for insightful discussions on improving KT and KT+ using centered kernels after the first version of this paper appeared on arXiv.

## Impact Statement

This paper presents work whose goal is to advance the field of Machine Learning. There are many potential societal consequences of our work, none which we feel must be

specifically highlighted here.

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

# Appendices

Appendix A describe further related work, including methods that employ non-uniform weights and foundational results on approximation complexity for numerical integration. The proofs for all theoretical results presented in the main text are contained in Appendix B. Finally, additional details required to reproduce our experiments are contained in Appendix C.

## A. Further Related Work

This appendix briefly summarises several strands of related work, mainly focusing on methods that employ non-uniform weights; an important distinction from the main text.

**Kernel Quadrature/Cubature:** There is a rich line of research in the machine learning literature on cubature rules based on integrating a minimal norm interpolant. Indeed, if we let $f_n$ denote the element of $\mathcal{H}$ that interpolates $f$ at each element of $\{\boldsymbol{x}_i\}_{i=1}^n$ and for which $\|f_n\|_{\mathcal{H}}$ is minimal, then $\int f_n \, \mathrm{d}\mu = \sum_{i=1}^n w_i^\star f(\boldsymbol{x}_i)$ takes the form of a weighted cubature rule. The performance of general weighted cubature rules $\sum_{i=1}^n w_i f(\boldsymbol{x}_i)$ can be analysed within the MMD framework by considering the associated weighted measure $\mu_n^{\boldsymbol{w}} = \sum_{i=1}^n w_i \delta_{\boldsymbol{x}_i}$. For *kernel cubature*[4], the weights $\boldsymbol{w}^\star$ minimise $\mathrm{MMD}(\mu, \mu_n^{\boldsymbol{w}})$ over all $\boldsymbol{w} \in \mathbb{R}^n$. In this case, using linearity of cubature rules,

$$\left| \sum_{i=1}^n w_i^\star f(\boldsymbol{x}_i) - \int f \, \mathrm{d}\mu \right|$$

$$= \left| \sum_{i=1}^n w_i^\star (f(\boldsymbol{x}_i) - f_n(\boldsymbol{x}_i)) - \int (f - f_n) \, \mathrm{d}\mu + \underbrace{\sum_{i=1}^n w_i^\star f_n(\boldsymbol{x}_i) - \int f_n \, \mathrm{d}\mu}_{=0} \right|$$

$$\leq \|f - f_n\|_{\mathcal{H}} \cdot \mathrm{MMD}(\mu, \mu_n^{\boldsymbol{w}^\star}),$$

where the final inequality is the same argument based on the reproducing property and Cauchy–Schwarz used to obtain (1) in the main text. Under weak conditions $\|f - f_n\|_{\mathcal{H}} \to 0$ as $n \to \infty$ [95, Theorem 2.1.1], which shows that the kernel cubature error is $o(\mathrm{MMD}(\mu, \mu_n^{\boldsymbol{w}^\star}))$, the same super-convergence property enjoyed by our stationary MMD points. Further, in this case one can often obtain the rate-optimal decay of MMD simply by taking $\boldsymbol{x}_i \sim \mu$ as independent [41, 67, 31], or as independent samples from a more appropriate task-aware sampling distribution [5, 25]. It is even possible to compute $\boldsymbol{w}^\star$ when $\mu$ is only implicitly defined [77] using Stein's method. However, the use of optimal weights comes at a considerable cost in terms of computation[5] and stability[6]; indeed, such estimators are rarely used in the numerical analysis community, where explicitly constructed cubatures are preferred.

**Determinantal Point Processes:** A determinantal point process (DPP) is a joint probability distribution on $\{\boldsymbol{x}_i\}_{i=1}^n$ defined via a kernel. For expositional simplicity here we discuss *projection* DPPs, for which the starting point is a kernel of the form

$$k(\boldsymbol{x}, \boldsymbol{x}') = \sum_{i=1}^n \psi_i(\boldsymbol{x}) \psi_i(\boldsymbol{x}') \tag{11}$$

where the $\psi_i$ are orthonormal functions in $L^2(\mu)$. Then

$$(\boldsymbol{x}_1, \ldots, \boldsymbol{x}_n) \mapsto \frac{1}{n!} \det[k(\boldsymbol{x}_i, \boldsymbol{x}_j)]_{i,j=1}^n \prod_{i=1}^n \mu(\boldsymbol{x}_i) \tag{12}$$

is a probability density [56, Lemma 21] whose $n$ marginals are all equal to $\mu$, while the components are in general correlated in a manner that depends on the kernel. Exact sampling from (12) is possible with access to the representation (11), while outside this case methods like rejection sampling are required; see the discussion in [45]. The integration error for a

---

[4]Kernel cubature is often called *Bayesian quadrature* due to an equivalent derivation in which $k$ is interpreted as the covariance function of a Gaussian process. This interpretation dates back at least to [68] and is regularly rediscovered; see the survey in [21].

[5]The general computational cost in exact arithmetic is $O(n^3)$. However, for specific targets $\mu$, kernels $k$, and point sets $\{\boldsymbol{x}_i\}_{i=1}^n$, several tricks are available to reduce this cost [see, e.g., 60, 63, 58].

[6]The stability of a weighted estimator is governed by the magnitude of the weights; if $|w_i|$ is large then small errors in computing $f(\boldsymbol{x}_i)$ are expounded. Kernel cubature is often unstable; see [62] for further detail.

continuously differentiable integrand $f$ with unweighted DPP points is $O(n^{-(1+1/d)/2})$ when $\mu$ is the $d$-fold product of so-called *Nevai class* distributions supported on $[-1, 1]$ [10, Theorem 2]. To overcome these strong restrictions on the form of $\mu$, DPPs can also be used along with importance sampling [10, Theorem 4]. One can further consider assigning kernel cubature weights to DPP points, and doing so recovers spectral rates [13]. The main limitation of DPPs relative to this work is that they are not able to produce uniformly weighted empirical approximations for general $\mu$ and efficient exact implementation only exists for Jacobi measures [45, 44].

**Information-Based Complexity:** The aim of *information-based complexity* is to analyse the theoretical difficulty of various approximation problems, and in particular results are available for numerical integration based on $n$ arbitrarily chosen evaluation locations $\{\boldsymbol{x}_i\}_{i=1}^n$. The $n$th *minimal error* in this context is defined as

$$e_k(n) \coloneqq \inf_{\boldsymbol{x}_1, \ldots, \boldsymbol{x}_n \in \mathcal{X}} \inf_{\boldsymbol{w} \in \mathbb{R}^n} \mathrm{MMD}(\mu, \mu_n^{\boldsymbol{w}}).$$

For certain kernels $k$ upper and lower bounds on $e_k(n)$ have been derived. For instance, [96, Proposition 5.3] established weak sufficient conditions on $k$ for which $e_k(n) = O((\log n)^{(5d+1)/2}/n)$, meaning that there exist numerical integration routines that are superior to Monte Carlo in the sense of MMD. Under stronger assumptions, for example that $\mathcal{H}$ is norm-equivalent to the *Sobolev space* $H^s([0, 1]^d)$, it is known that $e_k(n) \asymp n^{-s/d}$, so that the integration error for $f \in \mathcal{H}$ can converge arbitrarily fast depending on the smoothness of the kernel [76, Section 1.3.12, Proposition 3]. Further, for certain kernels (asymptotically) optimal nodes can be deduced [e.g., for the Sobolev case, space-filling nodes can be sufficient; 21]. These results are typically non-constructive and concern (optimally) weighted estimators. It is therefore remarkable that identical rates (up to logarithmic factors) can sometimes be achieved with explicit estimators that are uniformly weighted, such as QMC methods.

**Other Works:** Here we briefly mention other related works that do not fall neatly into the above categories. For kernel cubature, Oettershagen [78, Section 5.1] developed an efficient Newton method for optimising the locations of the nodes $\{\boldsymbol{x}_i\}_{i=1}^n$, but this used a strategy that applies only to dimension $d = 1$. Computation of the optimal weights $\boldsymbol{w}^\star$ in kernel cubature is associated with a $O(n^3)$ cost, and this motivates consideration of alternative weights that are easier to compute. Weights arising from Riemann sums and kernel density estimation were considered, respectively, in [80] and [35] in conjunction with independent samples $\boldsymbol{x}_i \sim \mu$. Weights arising from a nearest neighbour control variate method, in lieu of using a kernel, were considered in [70]. In each case the integration error is provably $o(n^{-1/2})$ for a sufficiently smooth integrand. Stratified sampling combined with finite difference approximation leads to a weighted estimator that is *polynomially exact*[7] and was shown to be optimal in the case of integrands $f \in C^r([0, 1]^d)$ in [34], the space of functions whose partial derivatives of order $\leq r$ exist and are continuous, The *recombination* method of [50] first generates a large number of independent samples from $\mu$ and then selects from these a sub-sample, based on which a polynomially exact weighted cubature rule is constructed, obtaining a integration error that can be related to the smoothness of the kernel. Unlike our stationary MMD points which are simulated via MMD Wasserstein gradient flow, both the weights and nodes are optimised via MMD Wasserstein Fisher–Rao gradient flow in [14]. See Table 1 in [25] for a more detailed comparison of several of the methods that we have mentioned.

# B. Proof of Theoretical Results

This appendix contains proofs for all theoretical results stated in the main text. First, in Appendix B.1, we present some preliminary results which will be useful. The statement and proof of Lemma B.6 are contained in Appendix B.3. The statement and proof of Lemma B.5 are contained in Appendix B.2. Finally the proof of Theorem 4.1 is contained in Appendix B.4.

## B.1. Preliminary Results

In this appendix several preliminary results are presented. The first result proves that one is allowed to interchange integral and derivative in (4) (Appendix B.1.1). The second result concerns the RKHS spanned by $k(\boldsymbol{x}, \cdot)$ for $\boldsymbol{x} \in \mathcal{X}$, and how this relates to the more familiar *restriction* of an RKHS to a sub-domain $\mathcal{X}$ (Appendix B.1.2), clarifying a technical aspect of our analysis. The third result is a general result stating conditions under which convergence $\mathrm{MMD}(\mu, \mu_n) \to 0$ implies denseness of the support points of the empirical measure $\mu_n$ in $\mathcal{X}$ (Appendix B.1.3).

---

[7]A cubature rule is *polynomially exact* if it is exact on all polynomials up to a specified finite degree. It is also possible to construct kernel cubature rules that are exact on a certain finite-dimensional linear subspace; see [61, 12].

### B.1.1. INTERCHANGE INTEGRALS AND DERIVATIVES

We begin with the interchange of differentiation and integration in (4).

**Lemma B.1.** *Suppose that the kernel $k$ satisfies Assumption 1. Then, $\nabla_{\boldsymbol{x}} \int k(\boldsymbol{x}, \boldsymbol{y}) \, \mathrm{d}\mu(\boldsymbol{y}) = \int \nabla_1 k(\boldsymbol{x}, \boldsymbol{y}) \, \mathrm{d}\mu(\boldsymbol{y})$ for any $\boldsymbol{x} \in \mathbb{R}^d$.*

*Proof.* The proof follows from Theorem A.7 in [37] by checking that (A.2a), (A.2b) and (A.8) in there are satisfied. Since $\sup_{\boldsymbol{x} \in \mathbb{R}^d} \int k(\boldsymbol{x}, \boldsymbol{y}) \, \mathrm{d}\mu(\boldsymbol{y}) < \infty$, (A.2a) is satisfied. Since for any $\ell \in \{1, \ldots, d\}$, $\sup_{\boldsymbol{x} \in \mathbb{R}^d} \int \partial_\ell k(\boldsymbol{x}, \boldsymbol{y}) \, \mathrm{d}\mu(\boldsymbol{y}) < \infty$, (A.2b) is satisfied. The derivative $(\boldsymbol{x}, \boldsymbol{y}) \mapsto \partial_\ell \partial_{\ell+d} k(\boldsymbol{x}, \boldsymbol{y})$ exists everywhere, so $(\boldsymbol{x}, \boldsymbol{y}) \mapsto k(\boldsymbol{x}, \boldsymbol{y})$ is continuous and consequently $k(\boldsymbol{x}, \cdot)$ is continuous [91, Lemma 4.29]. Denote $\boldsymbol{e}_\ell = [0, \ldots, 1, \ldots, 0]^\top$ as the unit vector whose $\ell$-th element is 1. Hence, from the mean value theorem [81], we know that there exists $\tilde{\boldsymbol{x}} \in \mathbb{R}^d$ such that

$$\frac{\Delta_{h,\ell} k(\boldsymbol{y}, \boldsymbol{x})}{h} := \frac{1}{h} \left( k\left(\boldsymbol{y}, \boldsymbol{x} + h\boldsymbol{e}_\ell\right) - k\left(\boldsymbol{y}, \boldsymbol{x}\right) \right) = \partial_{\ell+d} k\left(\boldsymbol{y}, \tilde{\boldsymbol{x}}\right).$$

Since $\boldsymbol{y} \mapsto h^{-1} \sup_{\boldsymbol{x} \in \mathbb{R}^d} \Delta_{h,\ell} k(\boldsymbol{y}, \boldsymbol{x})$ is $\mu$-integrable, (A.8) is satisfied. $\qquad\square$

### B.1.2. ON THE SUPPORT $\mathcal{X}$ AND THE RKHS SUBSET $\mathcal{H}_{\mathcal{X}}$

For the super-convergence result (Theorem 3.4), we restrict the integrand to functions in $\mathcal{H}_{\mathcal{X}}$ (up to an additive constant), which is defined as the closure $\overline{\mathrm{span}}\{k(\boldsymbol{x}, \cdot) : \boldsymbol{x} \in \mathcal{X}\}$. This appendix provides a rationale for this consideration (see Remark B.3). To this end, define the restriction map $V : (\mathbb{R}^d \to \mathbb{R}) \to (\mathcal{X} \to \mathbb{R})$ by $Vf(x) = f(x)$ for any $x \in \mathcal{X}$ and $f : \mathbb{R}^d \to \mathbb{R}$. We also introduce an RKHS $\mathcal{H}_{|\mathcal{X}}$ of functions on $\mathcal{X}$, which is defined by the reproducing kernel $k$ restricted to $\mathcal{X} \times \mathcal{X}$. Lemma B.2 expresses $\mathcal{H}_{\mathcal{X}}$ in terms of the restriction map $V$.

**Lemma B.2** (Relation between $\mathcal{H}_{\mathcal{X}}$ and $V$). *Let $k : \mathbb{R}^d \times \mathbb{R}^d \to \mathbb{R}$ be a positive semi-definite kernel and $\mathcal{H}$ be the corresponding RKHS. Let $\mathcal{X} \subseteq \mathbb{R}^d$ and $V$ be the associated restriction map on $\mathcal{H}$. We have $\mathcal{H}_{\mathcal{X}} = \mathcal{N}(V)^\perp$, where $\mathcal{N}(V)$ denotes the null space of $V$. Moreover, $V$ is an isometry between $\mathcal{H}_{\mathcal{X}}$ and $\mathcal{H}_{|\mathcal{X}}$.*

*Proof.* Both claims are consequences of [79, Corollary 5.8 and the surrounding discussion]. For completeness, we provide a proof for the first claim. For any $f \in \mathcal{H}_{\mathcal{X}}^\perp$, we have $(Vf)(\boldsymbol{x}) = f(\boldsymbol{x}) = \langle f, k(\boldsymbol{x}, \cdot) \rangle_{\mathcal{H}} = 0$ for all $\boldsymbol{x} \in \mathcal{X}$, so that $Vf \equiv 0$ and thus $\mathcal{H}_{\mathcal{X}}^\perp \subseteq \mathcal{N}(V)$, which implies $\mathcal{N}(V)^\perp \subseteq \mathcal{H}_{\mathcal{X}}$. On the other hand, let $h = \sum_{i=1}^n \alpha_i k(\boldsymbol{x}_i^{(t)}, \cdot) \in \mathrm{span}\{k(\boldsymbol{x}, \cdot) : \boldsymbol{x} \in \mathcal{X}\}$. For any $g \in \mathcal{N}(V)$, we have $\langle h, g \rangle_{\mathcal{H}} = \sum_{i=1}^n \alpha_i g(\boldsymbol{x}_i^{(t)}) = 0$, so that $h \in \mathcal{N}(V)^\perp$. Since $\mathcal{H}_{\mathcal{X}}$ is the closure of such finite linear combinations and $\mathcal{N}(V)^\perp$ is also closed, we have $\mathcal{H}_{\mathcal{X}} \subseteq \mathcal{N}(V)^\perp$. $\qquad\square$

**Remark B.3** (On the non-constant condition on $f \in \mathcal{H}_{\mathcal{X}}$). Proposition 3.3 and the subsequent Theorem 3.4 only concern functions in $\mathcal{H}_{\mathcal{X}}$ that are non-constant over $\mathcal{X}$. This restriction on the integrand only excludes functions with trivial integrals, which can be seen as follows: (a) since $\mathcal{H}_{\mathcal{X}} = \mathcal{N}(V)^\perp$ by Lemma B.2, for any $f \in \mathcal{H} \setminus \mathcal{H}_{\mathcal{X}} = \mathcal{N}(V) \cup \{0\}$, we have $\int f \mathrm{d}\mu = 0$ since $f(\boldsymbol{x}) = 0$ for all $\boldsymbol{x} \in \mathcal{X}$; (b) if $f \in \mathcal{H}_{\mathcal{X}}$ is constant on $\mathcal{X}$, i.e., $f \equiv c$ on $\mathcal{X}$ for some $c \in \mathbb{R}$, we have $\int f \mathrm{d}\mu = c$. The existence of such a constant-on-$\mathcal{X}$ function is equivalent to $\mathcal{H}_{|\mathcal{X}}$ having an everywhere constant function. Certain RKHSs do not have such constant functions; for example, the Gaussian RKHS does not have a constant function if $\mathcal{X}$ has a non-empty interior [91, Corollary 4.44]. For such an RKHS, the non-constancy requirement is unnecessary.

### B.1.3. DENSENESS OF STATIONARY MMD POINTS

**Lemma B.4** (Stationary MMD points are dense in $\mathcal{X}$). *Let $\mu \in \mathcal{P}(\mathbb{R}^d)$ and $\mathcal{X}$ be its support. Suppose the kernel $k$ satisfies Assumption 3. Let $(\mu_n)_{n \in \mathbb{N}}$ be a sequence of empirical measures with $\mu_n = n^{-1} \sum_{i=1}^n \delta_{\boldsymbol{x}_i^{(t)}}$ for some $\boldsymbol{x}_i^{(t)} \in \mathbb{R}^d$ and suppose that $\mathrm{MMD}(\mu, \mu_n) \to 0$ as $n \to \infty$. Then for each $\boldsymbol{x} \in \mathcal{X}$ and for any $\varepsilon > 0$, there exists $N_{\varepsilon, \boldsymbol{x}}$ such that $\min_{i=1,\ldots,n} \|\boldsymbol{x} - \boldsymbol{x}_i^{(t)}\| \leq \varepsilon$ for every $n > N_{\varepsilon, \boldsymbol{x}}$.*

*Proof of Lemma B.4.* Let $\boldsymbol{x} \in \mathcal{X}$ and $\varepsilon > 0$. Let $B_{2\varepsilon}(\boldsymbol{x}) := \{\boldsymbol{y} \in \mathbb{R}^d : \|\boldsymbol{y} - \boldsymbol{x}\| < 2\varepsilon\}$ denote a ball of radius $2\varepsilon$ about $\boldsymbol{x}$. The aim in this proof is to show that $\{\boldsymbol{x}_i^{(t)}\}_{i=1}^n \cap B_{2\varepsilon}(\boldsymbol{x}) \neq \emptyset$ for $n$ large enough.

By definition of the support, $\mu(B_\varepsilon(\boldsymbol{x})) > 0$. Let $f \in C_0(\mathbb{R}^d)$ be such that $f(\boldsymbol{x}) \in [0, 1]$ for all $\boldsymbol{x}$, $f(\boldsymbol{x}) = 1$ for $\boldsymbol{x} \in B_\varepsilon(\boldsymbol{x})$ and $f(\boldsymbol{x}) = 0$ for $\boldsymbol{x} \in \mathbb{R}^d \setminus B_{2\varepsilon}(\boldsymbol{x})$; see Lemma 2.22 in [69] for the existence of such a function. By the $C_0$-universality of

the RKHS $\mathcal{H}$, we have an element $f_\ell \in \mathcal{H}$ such that $\|f - f_\ell\|_\infty \leq \ell$, where we choose

$$\ell = \frac{\mu(B_\varepsilon(\boldsymbol{x}))}{6}.$$

Let $N_{\varepsilon, \boldsymbol{x}}$ be large enough that the inequality

$$\mathrm{MMD}(\mu, \mu_n) < \frac{\mu(B_\varepsilon(\boldsymbol{x}))}{6\|f_\ell\|_\mathcal{H}}.$$

holds for all $n \geq N_{\varepsilon, \boldsymbol{x}}$. Then for all such $n \geq N_{\varepsilon, \boldsymbol{x}}$ we have

$$\left| \int f \, \mathrm{d}(\mu_n - \mu) \right| \leq \int |f - f_\ell| \, \mathrm{d}(\mu_n + \mu) + \left| \int f_\ell \, \mathrm{d}(\mu_n - \mu) \right|$$

$$\leq 2\ell + \|f_\ell\|_\mathcal{H} \, \mathrm{MMD}(\mu, \mu_n)$$

$$\leq \frac{2}{6}\mu(B_\varepsilon(\boldsymbol{x})) + \frac{1}{6}\mu(B_\varepsilon(\boldsymbol{x}))$$

$$= \frac{1}{2}\mu(B_\varepsilon(\boldsymbol{x})).$$

Let $1_{B_{2\varepsilon}(\boldsymbol{x})}(\boldsymbol{y})$ denote the indicator function of the event $\boldsymbol{y} \in B_{2\varepsilon}(\boldsymbol{x})$. By construction, $f \leq 1_{B_{2\varepsilon}}$ on $\mathbb{R}^d$. From the reverse triangle inequality, the measure $\mu_n$ satisfies

$$\mu_n(B_{2\varepsilon}(\boldsymbol{x})) = \int 1_{B_{2\varepsilon}(\boldsymbol{x})}(\boldsymbol{y}) \, \mathrm{d}\mu_n(\boldsymbol{y}) \geq \int f \, \mathrm{d}\mu_n$$

$$\geq \left| \underbrace{\int f \, \mathrm{d}\mu}_{\geq \mu(B_\varepsilon(\boldsymbol{x}))} - \underbrace{\left| \int f \, \mathrm{d}(\mu_n - \mu) \right|}_{\leq \frac{1}{2}\mu(B_\varepsilon(\boldsymbol{x}))} \right| \geq \frac{1}{2}\mu(B_\varepsilon(\boldsymbol{x})) > 0,$$

meaning that $\{\boldsymbol{x}_i^{(t)}\}_{i=1}^n$ must contain at least one point in $B_{2\varepsilon}(\boldsymbol{x})$ for all $n \geq N_{\varepsilon, \boldsymbol{x}}$. $\qquad\square$

## B.2. Statement and Proof of Lemma B.5

Lemma B.5 is an important ingredient in the proof of Proposition 3.3. Recall $\mathcal{G}_\mathcal{X} = \mathrm{span}\{\partial_\ell k(\boldsymbol{x}, \cdot) : \boldsymbol{x} \in \mathcal{X}, 1 \leq \ell \leq d\}$.

**Lemma B.5** ($\mathcal{G}_\mathcal{X}$ is dense in $\mathcal{H}_\mathcal{X}$)**.** *Suppose $\mathcal{X}$ satisfies Assumption 2. Suppose the kernel $k$ satisfies Assumption 1. Fix $\varepsilon > 0$. For any function $f \in \mathcal{H}_\mathcal{X} : \mathbb{R}^d \to \mathbb{R}$ that is not constant on $\mathcal{X}$, there exists $g_\varepsilon \in \mathcal{G}_\mathcal{X}$ such that $\|f - g_\varepsilon\|_\mathcal{H} \leq \varepsilon$.*

*Proof of Lemma B.5.* Let us denote by $\mathcal{S} = \{f \in \mathcal{H}_\mathcal{X} : f_{|\mathcal{X}} \not\equiv \mathrm{const}\} \cup \{0\}$ the subset of functions in $\mathcal{H}_\mathcal{X}$ whose restrictions to $\mathcal{X}$ are not identically constant. We show that $\mathcal{G}_\mathcal{X}$ is dense relative to $\mathcal{S}$. Recall that a linear subspace of a Hilbert space is dense if and only if it has a trivial orthogonal complement. It thus suffices to prove that, for $f \in \mathcal{S}$, if $\langle \partial_\ell k(\boldsymbol{x}, \cdot), f \rangle_\mathcal{H} = \partial_\ell f(\boldsymbol{x}) = 0$ for any $\boldsymbol{x} \in \mathcal{X}$ and any $1 \leq \ell \leq d$, then $f \equiv 0$. By Assumption 2, there is a continuously differentiable path $\gamma$ that connects any two points in $\mathcal{X}$. With the vanishing gradient $\nabla f$ everywhere on $\mathcal{X}$, the fundamental theorem of calculus applied to $f \circ \gamma$ implies that $f$ has the same value at any two points. Thus, having a zero gradient everywhere on $\mathcal{X}$ implies that $f$ is constant on $\mathcal{X}$, which only admits zero by the definition of $\mathcal{S}$, leading to $f \equiv 0$. $\qquad\square$

## B.3. Statement and Proof of Lemma B.6

Lemma B.6 is also an important ingredient in the proof of Proposition 3.3. Recall

$$\mathcal{G}_{\mathcal{X}_n} := \mathrm{span}\{\partial_\ell k(\boldsymbol{x}, \cdot) : \boldsymbol{x} \in \mathcal{X}_n, \ 1 \leq \ell \leq d\} \subset \mathcal{H}, \quad \mathcal{X}_n = \{\boldsymbol{x}_i\}_{i=1}^n \subset \mathbb{R}^d,$$

where $\{\boldsymbol{x}_i\}_{i=1}^n$ are the stationary MMD points.

**Lemma B.6** ($\mathcal{G}_{\mathcal{X}_n}$ is dense in $\mathcal{G}_\mathcal{X}$)**.** *Suppose $\mathcal{X}$ satisfies Assumption 2. Suppose the kernel $k$ satisfies Assumption 1 and Assumption 3. For each $\varepsilon > 0$ and $g \in \mathcal{G}_\mathcal{X}$, there is $n \in \mathbb{N}$ such that there exists $g_{n,\varepsilon} \in \mathcal{G}_{\mathcal{X}_n}$ satisfying $\|g - g_{n,\varepsilon}\|_\mathcal{H} \leq \varepsilon$.*

*Proof of Lemma B.6.* Let $g(\cdot) = \sum_{j=1}^{M} \sum_{\ell=1}^{d} w_{j\ell} \partial_\ell k(\boldsymbol{y}_j, \cdot) \in \mathcal{G}_{\mathcal{X}}$ where $\boldsymbol{y}_j \in \mathcal{X}$ and $M \in \mathbb{N}$. We aim to approximate $g$ using an element from $\mathcal{G}_{\mathcal{X}_n}$. From the reproducing property we have

$$\|\partial_\ell k(\boldsymbol{y}_j, \cdot) - \partial_\ell k(\boldsymbol{y}, \cdot)\|_{\mathcal{H}}^2 = \partial_\ell \partial_{\ell+d} k(\boldsymbol{y}_j, \boldsymbol{y}_j) - 2\partial_\ell \partial_{\ell+d} k(\boldsymbol{y}_j, \boldsymbol{y}) + \partial_\ell \partial_{\ell+d} k(\boldsymbol{y}, \boldsymbol{y})$$

for all $\boldsymbol{y} \in \mathbb{R}^d$. Note that the partial derivatives $(\boldsymbol{x}, \boldsymbol{y}) \mapsto \partial_\ell \partial_{\ell+d} k(\boldsymbol{x}, \boldsymbol{y})$ are continuous from Assumption 1. By the continuity and Lemma B.4 (with the **convergence** property of MMD stationary points), there exists $N_{\epsilon,j} \in \mathbb{N}$ such that for all $n \geq N_{\varepsilon,j}$, there exists an element $\tilde{\boldsymbol{y}}_j \in \{\boldsymbol{x}_i^{(t)}\}_{i=1}^n$ sufficiently close to $\boldsymbol{y}_j$ in the sense that

$$\|\partial_\ell k(\boldsymbol{y}_j, \cdot) - \partial_\ell k(\tilde{\boldsymbol{y}}_j, \cdot)\|_{\mathcal{H}} < \frac{\varepsilon}{\sum_{j=1}^{M} \sum_{\ell=1}^{d} |w_{j\ell}|}$$

holds simultaneously for every $\ell \in \{1, \dots, d\}$. Let $N_\varepsilon := \max_{j=1,\dots,M} N_{\varepsilon,j} \in \mathbb{N}$. The corresponding function

$$g_{n,\varepsilon}(\cdot) = \sum_{j=1}^{M} \sum_{\ell=1}^{d} w_{j\ell} \partial_\ell k(\tilde{\boldsymbol{y}}_j, \cdot) \in \mathcal{G}_{\mathcal{X}_n}$$

satisfies

$$\|g - g_{n,\varepsilon}\|_{\mathcal{H}} = \left\| \sum_{j=1}^{M} \sum_{\ell=1}^{d} w_{j\ell} \partial_\ell k(\boldsymbol{y}_j, \cdot) - \sum_{j=1}^{M} \sum_{\ell=1}^{d} w_{j\ell} \partial_\ell k(\tilde{\boldsymbol{y}}_j, \cdot) \right\|_{\mathcal{H}}$$

$$\leq \sum_{j=1}^{M} \sum_{\ell=1}^{d} |w_{j\ell}| \|\partial_\ell k(\boldsymbol{y}_j, \cdot) - \partial_\ell k(\tilde{\boldsymbol{y}}_j, \cdot)\|_{\mathcal{H}} < \varepsilon,$$

as required. $\qquad\square$

### B.4. Proof of Theorem 4.1

In the following, we denote a tuple of $n$-variables taking values in $\mathbb{R}^d$ by an underlined variable; for example, $\underline{\boldsymbol{x}} = [\boldsymbol{x}_1, \dots, \boldsymbol{x}_n] \in (\mathbb{R}^d)^n$.

Our proof is based on the following two claims proved in Appendix B.4.1 and Appendix B.4.2:

**Lemma B.7** (Descent lemma in expectation). *Suppose the kernel $k$ satisfies Assumption 1 and 4. Given initial particles $\{\boldsymbol{x}_i^{(1)}\}_{i=1}^n$, suppose the noisy MMD particle descent defined in* (8) *satisfies Assumption 5. Suppose step size $\gamma$ satisfies $256\gamma^2 d^2 \kappa \leq 1$ for any $t \in \mathbb{N}^+$. Then, we have*

$$\mathrm{MMD}^2 \left( \frac{1}{n} \sum_{i=1}^{n} \delta_{\boldsymbol{x}_i^{(t)}}, \mu \right) - \mathbb{E}_{\underline{\boldsymbol{u}}^{(t)}} \left[ \mathrm{MMD}^2 \left( \frac{1}{n} \sum_{i=1}^{n} \delta_{\boldsymbol{x}_i^{(t+1)}}, \mu \right) \right]$$

$$\geq \gamma \beta_t^2 \kappa d^2 \, \mathrm{MMD}^2 \left( \frac{1}{n} \sum_{i=1}^{n} \delta_{\boldsymbol{x}_i^{(t)}}, \mu \right). \tag{13}$$

**Lemma B.8** (Concentration of particles after noise injection). *Suppose the kernel $k$ satisfies Assumption 1 and 4. Then for any $t \in \{1, \dots, T-1\}$, given the particles $\{\boldsymbol{x}_i^{(t)}\}_{i=1}^n$, for any $\tau > 1$, we have with probability at least $1 - \exp(-\tau)$,*

$$\mathrm{MMD} \left( \frac{1}{n} \sum_{i=1}^{n} \delta_{\boldsymbol{x}_i^{(t+1)}}, \mu \right) - \mathbb{E}_{\underline{\boldsymbol{u}}^{(t)}} \left[ \mathrm{MMD} \left( \frac{1}{n} \sum_{i=1}^{n} \delta_{\boldsymbol{x}_i^{(t+1)}}, \mu \right) \right] \leq C\tau n^{-\frac{1}{2}} \left( (\gamma\beta_t)^{\frac{1}{2}} \vee n^{-\frac{1}{2}} \right) \sqrt{\kappa}\sqrt{d}. \tag{14}$$

*Here, $C$ is a positive universal constants independent of $n, \beta_t, \gamma, t$.*

**Remark B.9** (Descent lemma and concentration of particles). Lemma B.7 shows that, under the noise scaling condition in Assumption 5, the MMD flow trajectory satisfy a *gradient dominance* condition in expectation (also known as the Polyak–Lojasiewicz inequality), ensuring that the MMD decreases monotonically in expectation as desired. Lemma B.7 is commonly referred to as the 'descent lemma' in the context of convex optimisation. In Lemma B.8, conditioned on the particles $\underline{\boldsymbol{x}}^{(t)}$, the randomness in the next iteration $\underline{\boldsymbol{x}}^{(t+1)}$ arises solely from the injected Gaussian noise $\underline{\boldsymbol{u}}^{(t)}$ at iteration $t$. Accordingly, the probability is taken with respect to the joint distribution of $\underline{\boldsymbol{u}}^{(t)}$. Lemma B.8 then shows that, with high probability, the updated particles $\underline{\boldsymbol{x}}^{(t+1)}$ remain close to their expected values following noise injection.

*Proof of Theorem 4.1.* Assuming Lemma B.7 and Lemma B.8 hold, we can prove Theorem 4.1 as follows. Rearranging the terms in (13) and taking the square root yields,

$$\sqrt{\mathbb{E}_{\underline{\boldsymbol{u}}^{(t)}}\left[\mathrm{MMD}^2\left(\frac{1}{n}\sum_{i=1}^{n}\delta_{\boldsymbol{x}_i^{(t+1)}},\mu\right)\right]} \leq \left(1-\gamma\beta_t^2\kappa d^2\right)^{\frac{1}{2}}\mathrm{MMD}\left(\frac{1}{n}\sum_{i=1}^{n}\delta_{\boldsymbol{x}_i^{(t)}},\mu\right).$$

Then, we use Jensen's inequality and (14) to lower bound the left hand side of the above inequality. For any $\tau > 1$, with probability at least $1 - \exp(-\tau)$,

$$\sqrt{\mathbb{E}_{\underline{\boldsymbol{u}}^{(t)}}\left[\mathrm{MMD}^2\left(\frac{1}{n}\sum_{i=1}^{n}\delta_{\boldsymbol{x}_i^{(t+1)}},\mu\right)\right]} \geq \mathbb{E}_{\underline{\boldsymbol{u}}^{(t)}}\left[\mathrm{MMD}\left(\frac{1}{n}\sum_{i=1}^{n}\delta_{\boldsymbol{x}_i^{(t+1)}},\mu\right)\right]$$

$$\geq \mathrm{MMD}\left(\frac{1}{n}\sum_{i=1}^{n}\delta_{\boldsymbol{x}_i^{(t+1)}},\mu\right) - C\tau n^{-\frac{1}{2}}\left((\gamma\beta_t)^{\frac{1}{2}}\vee n^{-\frac{1}{2}}\right)\sqrt{\kappa}\sqrt{d}.$$

Here, $C$ is a positive universal constants independent of $n, \beta_t, t$. Therefore, with probability at least $1 - \exp(-\tau)$,

$$\mathrm{MMD}\left(\frac{1}{n}\sum_{i=1}^{n}\delta_{\boldsymbol{x}_i^{(t+1)}},\mu\right) \leq \left(1-\gamma\beta_t^2\kappa d^2\right)^{\frac{1}{2}}\mathrm{MMD}\left(\frac{1}{n}\sum_{i=1}^{n}\delta_{\boldsymbol{x}_i^{(t)}},\mu\right)$$

$$+ C\tau n^{-\frac{1}{2}}\left((\gamma\beta_t)^{\frac{1}{2}}\vee n^{-\frac{1}{2}}\right)\sqrt{\kappa}\sqrt{d},$$

for any $t \in \mathbb{N}$. Taking the union bound over all probabilities at time $t = 1, \ldots, T-1$ and applying the discrete Grönwall's lemma in Lemma B.12 yields the following: with probability at least $1 - (T-1)\exp(-\tau)$,

$$\mathrm{MMD}\left(\frac{1}{n}\sum_{i=1}^{n}\delta_{\boldsymbol{x}_i^{(T)}},\mu\right) \leq \prod_{t=1}^{T-1}\left(1-\gamma\beta_t^2\kappa d^2\right)^{\frac{1}{2}}\mathrm{MMD}\left(\frac{1}{n}\sum_{i=1}^{n}\delta_{\boldsymbol{x}_i^{(1)}},\mu\right)$$

$$+ n^{-\frac{1}{2}}\cdot\left(\sum_{t=1}^{T-1}C\tau\left((\gamma\beta_t)^{\frac{1}{2}}\vee n^{-\frac{1}{2}}\right)\sqrt{\kappa}\sqrt{d}\cdot\prod_{s=t+1}^{T-1}\left(1-\gamma\beta_s^2\kappa d^2\right)^{\frac{1}{2}}\right)$$

$$\leq \exp\left(-\sum_{t=1}^{T-1}\frac{1}{2}\gamma\beta_t^2\kappa d^2\right)\mathrm{MMD}\left(\frac{1}{n}\sum_{i=1}^{n}\delta_{\boldsymbol{x}_i^{(1)}},\mu\right)$$

$$+ n^{-\frac{1}{2}}\cdot\left(\sum_{t=1}^{T-1}C\tau\left((\gamma\beta_t)^{\frac{1}{2}}\vee n^{-\frac{1}{2}}\right)\sqrt{\kappa}\sqrt{d}\cdot\exp\left(-\sum_{s=t+1}^{T-1}\frac{1}{2}\gamma\beta_s^2\kappa d^2\right)\right).$$

The last inequality holds by using $1 - z \leq \exp(-z)$ for any $z > 0$. Finally, take $\tau = \delta + \log T$ for concludes the proof. $\square$

*Proof of Corollary 4.2.* Now we study the asymptotic behavior of the right hand side of (10) as $T \to \infty$ under the condition that $\beta_t \propto t^{-\alpha}$ for $0 \leq \alpha < 1/2$. Recall that the right hand side of (10) consists of optimisation error and estimation error. First, we analyse the optimisation error term.

$$\text{Optimisation error:} \quad \exp\left(-\frac{1}{2}\kappa d^2\sum_{t=1}^{T-1}\gamma\beta_t^2\right) \leq \exp\left(-\frac{1}{2}\kappa d^2\int_1^T\gamma\beta_t^2\mathrm{d}t\right) \asymp \exp\left(-\frac{1}{2}T^{1-2\alpha}\kappa d^2\right).$$

The first inequality holds because $\beta_t$ is decreasing in $t$, and the second asymptotic relation holds because $\beta_t \asymp t^{-\alpha}$. Next, we analyse the finite-sample estimation error term. Since $\beta_t^{1/2} \geq \beta_T^{1/2} \asymp T^{-\alpha/2} \geq n^{-\frac{1}{2}}$, so $(\beta_t\gamma)^{\frac{1}{2}}\vee n^{-\frac{1}{2}} = \beta_t$. Let $T_0 \in [1, T]$ to be decided later. Notice that

$$(*) := \sum_{t=1}^{T-1}\beta_t\cdot\exp\left(-\sum_{s=t+1}^{T-1}\frac{1}{2}\gamma\beta_s^2\kappa d^2\right)$$

$$= \sum_{t=1}^{T_0} \beta_t \cdot \exp\left(-\sum_{s=t+1}^{T-1} \frac{1}{2}\gamma\beta_s^2\kappa d^2\right) + \sum_{t=T_0+1}^{T-1} \beta_t \cdot \exp\left(-\sum_{s=t+1}^{T-1} \frac{1}{2}\gamma\beta_s^2\kappa d^2\right)$$

$$\leq \left(\sum_{t=1}^{T_0} t^{-\alpha}\right) \cdot \exp\left(-\frac{1}{2}(T-T_0)T^{-2\alpha}\kappa d^2\right) + \sum_{t=T_0+1}^{T-1} t^{-\alpha}$$

$$\lesssim T_0^{1-\alpha} \exp\left(-\frac{1}{2}(T-T_0)T^{-2\alpha}\kappa d^2\right) + (T-T_0)T_0^{-\alpha}.$$

Choose $T_0 = T - 2(\kappa d^2)^{-1}T^{2\alpha}(\log T)$ which is positive for $T$ sufficiently large since $2\alpha < 1$. Then, we obtain

$$(*) \lesssim (T - T^{2\alpha}\log T)^{1-\alpha}T^{-1} + (T^{2\alpha}\log T)(T - T^{2\alpha}\log T)^{-\alpha}$$
$$\lesssim T^{-\alpha} + T^\alpha \log T(1 - T^{2\alpha-1}\log T)^{-\alpha}$$
$$\lesssim T^\alpha \log T.$$

As a result, we have

$$\text{Estimation error} \lesssim n^{-\frac{1}{2}}(\log T)^2 T^\alpha.$$

Combining the above two error terms would finish the proof. $\qquad\square$

**Corollary B.10.** *In the setting of Theorem 4.1, suppose the initial particles $\{x_i^{(1)}\}_{i=1}^n$ satisfy $\mathrm{MMD}(\hat{\nu}_n^{(1)}, \mu) = \mathcal{O}(n^{-1}(\log n)^b)$ for some $b > 0$, suppose that $\gamma \cdot \beta_t = n^{-1}$ and that Assumption 5 holds. Then, for $T = \mathcal{O}((\log n)^b)$, $\mathrm{MMD}(\hat{\nu}_n^{(T)}, \mu) = \mathcal{O}(n^{-1}(\log n)^b)$.*

*Proof of Corollary B.10.* From Theorem 4.1, plug in the condition that $\mathrm{MMD}(\frac{1}{n}\sum_{i=1}^n \delta_{x_i^{(1)}}, \mu) = \mathcal{O}(n^{-1}(\log n)^b)$ and $\gamma\beta_t = n^{-1}$, then we have

$$\mathrm{MMD}\left(\frac{1}{n}\sum_{i=1}^n \delta_{x_i^{(T)}}, \mu\right) \lesssim \frac{(\log n)^b}{n} + \frac{\log T}{n} \cdot \left(\sum_{t=1}^{T-1} \exp\left(-\sum_{s=t+1}^{T-1} \frac{1}{2}\gamma\beta_s^2\kappa d^2\right)\right)$$
$$\lesssim \frac{(\log n)^b}{n} + \frac{\log T}{n}T.$$

Therefore, if we take $T = \mathcal{O}((\log n)^b)$, then we have $\mathrm{MMD}(\hat{\nu}_n^{(T)}, \mu) = \mathcal{O}(n^{-1}(\log n)^b)$ which concludes the proof. $\quad\square$

### B.4.1. PROOF OF LEMMA B.7

*Proof.* In the proof, we omit superscript and subscript $t$ for simplicity denoting $x_i^{(t)}, u_i^{(t)}, \beta_t$ as $x_i, u_i, \beta$. Define $\mathbf{T}_\tau\colon (\underline{x}, \underline{u}) \mapsto [T_{1,\tau}(\underline{x}, \underline{u}), \dots, T_{n,\tau}(\underline{x}, \underline{u})] \in (\mathbb{R}^d)^n$ with

$$T_{i,\tau}(\underline{x}, \underline{u}) = x_i + \tau\gamma\frac{1}{n}\sum_{j=1}^n \Phi(x_i + \beta_t u_i, x_j), \tag{15}$$

where $\Phi(z, w) = \mathbb{E}_{y\sim\mu}[\nabla_1 k(z, y)] - \nabla_1 k(z, w)$. By definition, the transformation satisfies $\underline{x} = \mathbf{T}_0(\underline{x}, \underline{u})$ for the current particles, and $\mathbf{T}_1(\underline{x}, \underline{u})$ represent the updated particles at the next iteration.

Define $E(\tau)$ as

$$E(\tau) := \mathbb{E}_{\underline{u}}\left[\mathrm{MMD}^2\left(\frac{1}{n}\sum_{i=1}^n \delta_{T_{i,1}(\underline{x},\underline{u})}, \mu\right)\right]$$
$$= \mathbb{E}_{\underline{u}}\left[\frac{1}{n^2}\sum_{i,j=1}^n k\left(T_{i,\tau}(\underline{x}, \underline{u}), T_{j,\tau}(\underline{x}, \underline{u})\right)\right]$$

$$- 2\,\mathbb{E}_{\underline{u}}\,\mathbb{E}_{\boldsymbol{y}\sim\mu}\left[\frac{1}{n}\sum_{i=1}^{n}k(T_{i,\tau}(\underline{\boldsymbol{x}},\underline{\boldsymbol{u}}),\boldsymbol{y})\right] + \text{const.}$$

Now to prove the lemma is equivalent to prove that $E(1) - E(0) \leq -\frac{1}{4}\gamma\beta^2\kappa d^2 E(0)$. Define $\Psi_{i,j}(\boldsymbol{u}_i) = \Phi(\boldsymbol{x}_i + \beta_t\boldsymbol{u}_i, \boldsymbol{x}_j)$ to help simplify the notation.

From Assumption 1, $k\colon \mathbb{R}^d \times \mathbb{R}^d \to \mathbb{R}$ has continuous derivatives with respect to both arguments, hence $E$ is differentiable with respect to $\tau$ and we are allowed to interchange derivative and integral by the Leibniz integral rule [81]. Then, we have

$$\dot{E}(\tau) := \frac{d}{d\tau}E(\tau)$$

$$= \mathbb{E}_{\underline{u}}\left[\frac{2}{n^2}\sum_{i,j=1}^{n}\nabla_1 k\left(T_{i,\tau}(\underline{\boldsymbol{x}},\underline{\boldsymbol{u}}),T_{j,\tau}(\underline{\boldsymbol{x}},\underline{\boldsymbol{u}})\right)^{\top}\left(\gamma\frac{1}{n}\sum_{\ell=1}^{n}\Psi_{i,\ell}(\boldsymbol{u}_i)\right)\right]$$

$$- 2\,\mathbb{E}_{\underline{u}}\,\mathbb{E}_{\boldsymbol{y}\sim\mu}\left[\frac{1}{n}\sum_{i=1}^{n}\nabla_1 k(T_{i,\tau}(\underline{\boldsymbol{x}},\underline{\boldsymbol{u}}),\boldsymbol{y})^{\top}\left(\gamma\frac{1}{n}\sum_{\ell=1}^{n}\Psi_{i,\ell}(\boldsymbol{u}_i)\right)\right]$$

$$= -2\gamma\,\mathbb{E}_{\underline{u}}\left[\frac{1}{n}\sum_{i=1}^{n}\left(\frac{1}{n}\sum_{j=1}^{n}\Phi(T_{i,\tau}(\underline{\boldsymbol{x}},\underline{\boldsymbol{u}}),T_{j,\tau}(\underline{\boldsymbol{x}},\underline{\boldsymbol{u}}))\right)^{\top}\left(\frac{1}{n}\sum_{\ell=1}^{n}\Psi_{i,\ell}(\boldsymbol{u}_i)\right)\right], \tag{16}$$

where in the last step we use the definition of $\Phi(\boldsymbol{z},\boldsymbol{w}) = \mathbb{E}_{\boldsymbol{y}\sim\mu}[\nabla_1 k(\boldsymbol{z},\boldsymbol{y})] - \nabla_1 k(\boldsymbol{z},\boldsymbol{w})$. Hence, since $\underline{\boldsymbol{x}} = \mathbf{T}_0(\underline{\boldsymbol{x}},\underline{\boldsymbol{u}})$,

$$\dot{E}(0) = -2\gamma\,\mathbb{E}_{\underline{u}}\left[\frac{1}{n}\sum_{i=1}^{n}\left(\frac{1}{n}\sum_{j=1}^{n}\Phi(\boldsymbol{x}_i,\boldsymbol{x}_j)\right)^{\top}\left(\frac{1}{n}\sum_{\ell=1}^{n}\Psi_{i,\ell}(\boldsymbol{u}_i)\right)\right]. \tag{17}$$

We can further upper bound $\dot{E}(0)$ as

$$\dot{E}(0) = -2\gamma\,\mathbb{E}_{\underline{u}}\left[\frac{1}{n}\sum_{i=1}^{n}\left(\frac{1}{n}\sum_{j=1}^{n}\Psi_{i,j}(\boldsymbol{u}_i)\right)^{\top}\left(\frac{1}{n}\sum_{\ell=1}^{n}\Psi_{i,\ell}(\boldsymbol{u}_i)\right)\right]$$

$$- 2\gamma\,\mathbb{E}_{\underline{u}}\left[\frac{1}{n}\sum_{i=1}^{n}\left(\frac{1}{n}\sum_{j=1}^{n}\Phi(\boldsymbol{x}_i,\boldsymbol{x}_j) - \frac{1}{n}\sum_{j=1}^{n}\Psi_{i,j}(\boldsymbol{u}_i)\right)^{\top}\left(\frac{1}{n}\sum_{\ell=1}^{n}\Psi_{i,\ell}(\boldsymbol{u}_i)\right)\right]$$

$$\overset{(*)}{\leq} -2\gamma\,\mathbb{E}_{\underline{u}}\left[\frac{1}{n}\sum_{i=1}^{n}\left\|\frac{1}{n}\sum_{j=1}^{n}\Psi_{i,j}(\boldsymbol{u}_i)\right\|^2\right] + \gamma\,\mathbb{E}_{\underline{u}}\left[\frac{1}{n}\sum_{i=1}^{n}\left\|\frac{1}{n}\sum_{j=1}^{n}\Psi_{i,j}(\boldsymbol{u}_i)\right\|^2\right]$$

$$+ \gamma\,\mathbb{E}_{\underline{u}}\left[\frac{1}{n}\sum_{i=1}^{n}\left\|\frac{1}{n}\sum_{j=1}^{n}\Phi(\boldsymbol{x}_i,\boldsymbol{x}_j) - \frac{1}{n}\sum_{j=1}^{n}\Psi_{i,j}(\boldsymbol{u}_i)\right\|^2\right]$$

$$= -\gamma\,\mathbb{E}_{\underline{u}}\left[\frac{1}{n}\sum_{i=1}^{n}\left\|\frac{1}{n}\sum_{j=1}^{n}\Psi_{i,j}(\boldsymbol{u}_i)\right\|^2\right]$$

$$+ \gamma\,\mathbb{E}_{\underline{u}}\left[\frac{1}{n}\sum_{i=1}^{n}\left\|\frac{1}{n}\sum_{j=1}^{n}\Phi(\boldsymbol{x}_i,\boldsymbol{x}_j) - \frac{1}{n}\sum_{j=1}^{n}\Psi_{i,j}(\boldsymbol{u}_i)\right\|^2\right].$$

In $(*)$, we use $-a^{\top}b \leq \frac{1}{2}\|a\|^2 + \frac{1}{2}\|b\|^2$ for any $a, b \in \mathbb{R}^d$ for the second term. By using Lemma B.11, we have

$$\mathbb{E}_{\underline{u}}\left[\left\|\frac{1}{n}\sum_{j=1}^{n}\Phi(\boldsymbol{x}_i,\boldsymbol{x}_j) - \frac{1}{n}\sum_{j=1}^{n}\Psi_{i,j}(\boldsymbol{u}_i)\right\|\right] \leq d\kappa\beta\,\mathbb{E}_{\underline{u}}[\|\boldsymbol{u}_i\|]\,\text{MMD}\left(\frac{1}{n}\sum_{j=1}^{n}\delta_{\boldsymbol{x}_j},\mu\right)$$

$$\leq d^2 \kappa \beta \operatorname{MMD} \left( \frac{1}{n} \sum_{j=1}^{n} \delta_{\boldsymbol{x}_j}, \mu \right).$$

It follows that, by using Assumption 5,

$$\dot{E}(0) \leq -\gamma \mathbb{E}_{\underline{\boldsymbol{u}}} \left[ \frac{1}{n} \sum_{i=1}^{n} \left\| \frac{1}{n} \sum_{j=1}^{n} \Psi_{i,j}(\boldsymbol{u}_i) \right\|^2 \right] + \gamma d^2 \kappa \beta^2 \operatorname{MMD}^2 \left( \frac{1}{n} \sum_{j=1}^{n} \delta_{\boldsymbol{x}_j}, \mu \right)$$

$$\leq -\frac{3}{4} \gamma \mathbb{E}_{\underline{\boldsymbol{u}}} \left[ \frac{1}{n} \sum_{i=1}^{n} \left\| \frac{1}{n} \sum_{j=1}^{n} \Psi_{i,j}(\boldsymbol{u}_i) \right\|^2 \right]. \tag{18}$$

On the other hand, combine both (17) and (16), we also have

$$\left| \dot{E}(\tau) - \dot{E}(0) \right|$$

$$= \left| 2\gamma \mathbb{E}_{\underline{\boldsymbol{u}}} \left[ \frac{1}{n} \sum_{i=1}^{n} \left( \frac{1}{n} \sum_{j=1}^{n} \Phi(T_{i,\tau}(\underline{\boldsymbol{x}}, \underline{\boldsymbol{u}}), T_{j,\tau}(\underline{\boldsymbol{x}}, \underline{\boldsymbol{u}})) \right)^{\top} \left( \frac{1}{n} \sum_{\ell=1}^{n} \Psi_{i,\ell}(\boldsymbol{u}_i) \right) \right] \right.$$

$$\left. - 2\gamma \mathbb{E}_{\underline{\boldsymbol{u}}} \left[ \frac{1}{n} \sum_{i=1}^{n} \left( \frac{1}{n} \sum_{j=1}^{n} \Phi(\boldsymbol{x}_i, \boldsymbol{x}_j) \right)^{\top} \left( \frac{1}{n} \sum_{\ell=1}^{n} \Psi_{i,\ell}(\boldsymbol{u}_i) \right) \right] \right|$$

$$= \left| 2\gamma \mathbb{E}_{\underline{\boldsymbol{u}}} \left[ \frac{1}{n} \sum_{i=1}^{n} \left( \frac{1}{n} \sum_{j=1}^{n} \Phi(T_{i,\tau}(\underline{\boldsymbol{x}}, \underline{\boldsymbol{u}}), T_{j,\tau}(\underline{\boldsymbol{x}}, \underline{\boldsymbol{u}})) - \frac{1}{n} \sum_{j=1}^{n} \Phi(\boldsymbol{x}_i, \boldsymbol{x}_j) \right)^{\top} \left( \frac{1}{n} \sum_{\ell=1}^{n} \Psi_{i,\ell}(\boldsymbol{u}_i) \right) \right] \right|$$

$$\overset{(*)}{\leq} 4\gamma \mathbb{E}_{\underline{\boldsymbol{u}}} \left[ \frac{1}{n} \sum_{i=1}^{n} \left\| \frac{1}{n} \sum_{j=1}^{n} \Phi(T_{i,\tau}(\underline{\boldsymbol{x}}, \underline{\boldsymbol{u}}), T_{j,\tau}(\underline{\boldsymbol{x}}, \underline{\boldsymbol{u}})) - \frac{1}{n} \sum_{j=1}^{n} \Phi(\boldsymbol{x}_i, \boldsymbol{x}_j) \right\|^2 \right]$$

$$+ \frac{1}{4} \gamma \mathbb{E}_{\underline{\boldsymbol{u}}} \left[ \frac{1}{n} \sum_{i=1}^{n} \left\| \frac{1}{n} \sum_{\ell=1}^{n} \Psi_{i,\ell}(\boldsymbol{u}_i) \right\|^2 \right]. \tag{19}$$

In $(*)$, we use $a^{\top}b \leq \frac{1}{8}\|a\|^2 + 2\|b\|^2$ for any $a, b \in \mathbb{R}^d$. Note that Lemma B.11 implies

$$\| \Phi(T_{i,\tau}(\underline{\boldsymbol{x}}, \underline{\boldsymbol{u}}), T_{j,\tau}(\underline{\boldsymbol{x}}, \underline{\boldsymbol{u}})) - \Phi(\boldsymbol{x}_i, \boldsymbol{x}_j) \|$$

$$\leq 2d\kappa\tau\gamma \left( \left\| \frac{1}{n} \sum_{\ell=1}^{n} \Phi(\boldsymbol{x}_i + \beta_t \boldsymbol{u}_i, \boldsymbol{x}_\ell) \right\| + \left\| \frac{1}{n} \sum_{\ell=1}^{n} \Phi(\boldsymbol{x}_j + \beta_t \boldsymbol{u}_j, \boldsymbol{x}_\ell) \right\| \right). \tag{20}$$

By (20), we notice that

$$\| \Phi(T_{i,\tau}(\underline{\boldsymbol{x}}, \underline{\boldsymbol{u}}), T_{j,\tau}(\underline{\boldsymbol{x}}, \underline{\boldsymbol{u}})) - \Phi(\boldsymbol{x}_i, \boldsymbol{x}_j) \|^2$$

$$\leq 4d^2 \kappa \tau^2 \gamma^2 \left( \left\| \frac{1}{n} \sum_{\ell=1}^{n} \Phi(\boldsymbol{x}_i + \beta \boldsymbol{u}_i, \boldsymbol{x}_\ell) \right\| + \left\| \frac{1}{n} \sum_{\ell=1}^{n} \Phi(\boldsymbol{x}_j + \beta \boldsymbol{u}_j, \boldsymbol{x}_\ell) \right\| \right)^2$$

$$\leq 8d^2 \kappa \tau^2 \gamma^2 \left( \left\| \frac{1}{n} \sum_{\ell=1}^{n} \Psi_{i,\ell}(\boldsymbol{u}_i) \right\|^2 + \left\| \frac{1}{n} \sum_{\ell=1}^{n} \Psi_{j,\ell}(\boldsymbol{u}_j) \right\|^2 \right).$$

So we continue from (19) to obtain

$$\left| \dot{E}(\tau) - \dot{E}(0) \right|$$

$$\leq 4\gamma\, \mathbb{E}_{\underline{\boldsymbol{u}}} \left[ \frac{1}{n^2} \sum_{i,j=1}^{n} 8d^2\kappa\tau^2\gamma^2 \left( \left\| \frac{1}{n} \sum_{\ell=1}^{n} \Psi_{i,\ell}(\boldsymbol{u}_i) \right\|^2 + \left\| \frac{1}{n} \sum_{\ell=1}^{n} \Psi_{j,\ell}(\boldsymbol{u}_j) \right\|^2 \right) \right]$$

$$+ \frac{1}{4}\gamma\, \mathbb{E}_{\underline{\boldsymbol{u}}} \left[ \frac{1}{n} \sum_{i=1}^{n} \left\| \frac{1}{n} \sum_{\ell=1}^{n} \Psi_{i,\ell}(\boldsymbol{u}_i) \right\|^2 \right]$$

$$= 64d^2\kappa\tau^2\gamma^3\, \mathbb{E}_{\underline{\boldsymbol{u}}} \left[ \frac{1}{n} \sum_{i=1}^{n} \left\| \frac{1}{n} \sum_{\ell=1}^{n} \Psi_{i,\ell}(\boldsymbol{u}_i) \right\|^2 \right] + \frac{1}{4}\gamma\, \mathbb{E}_{\underline{\boldsymbol{u}}} \left[ \frac{1}{n} \sum_{i=1}^{n} \left\| \frac{1}{n} \sum_{\ell=1}^{n} \Psi_{i,\ell}(\boldsymbol{u}_i) \right\|^2 \right]$$

$$\leq \frac{1}{2}\gamma\, \mathbb{E}_{\underline{\boldsymbol{u}}} \left[ \frac{1}{n} \sum_{i=1}^{n} \left\| \frac{1}{n} \sum_{\ell=1}^{n} \Psi_{i,\ell}(\boldsymbol{u}_i) \right\|^2 \right]. \tag{21}$$

In the above derivations, the last step holds because of the condition that $256\gamma^2 d^2\kappa \leq 1$. Hence, combining (18) and (21), and since $E$ is differentiable with respect to $\tau$, we obtain that

$$E(1) - E(0) = \dot{E}(0) + \int_0^1 \left( \dot{E}(\tau) - \dot{E}(0) \right) d\tau$$

$$\leq -\frac{3}{4}\gamma\, \mathbb{E}_{\underline{\boldsymbol{u}}} \left[ \frac{1}{n} \sum_{i=1}^{n} \left\| \frac{1}{n} \sum_{j=1}^{n} \Psi_{i,j}(\boldsymbol{u}_i) \right\|^2 \right] + \int_0^1 \frac{1}{2}\gamma\, \mathbb{E}_{\underline{\boldsymbol{u}}} \left[ \frac{1}{n} \sum_{i=1}^{n} \left\| \frac{1}{n} \sum_{\ell=1}^{n} \Psi_{i,\ell}(\boldsymbol{u}_i) \right\|^2 \right] d\tau$$

$$= -\frac{\gamma}{4}\, \mathbb{E}_{\underline{\boldsymbol{u}}} \left[ \frac{1}{n} \sum_{i=1}^{n} \left\| \frac{1}{n} \sum_{j=1}^{n} \Psi_{i,j}(\boldsymbol{u}_i) \right\|^2 \right]$$

$$\leq -\gamma\beta^2\kappa d^2\, \mathrm{MMD}^2 \left( \frac{1}{n} \sum_{i=1}^{n} \delta_{\boldsymbol{x}_i^{(t)}}, \mu \right) = -\gamma\beta^2\kappa d^2 E(0).$$

Therefore, we have proved that $E(1) - E(0) \leq -\gamma\beta^2\kappa d^2 E(0)$ which concludes the proof. $\qquad\square$

### B.4.2. PROOF OF LEMMA B.8

*Proof.* Recall that at iteration $t$,

$$\boldsymbol{x}_i^{(t+1)} = \boldsymbol{x}_i^{(t)} - \gamma \left\{ \mathbb{E}_{\boldsymbol{Y}\sim\mu}[\nabla_1 k(\boldsymbol{x}_i^{(t)} + \beta_t\boldsymbol{u}_i^{(t)}, \boldsymbol{Y})] - \frac{1}{n} \sum_{j=1}^{n} \nabla_1 k(\boldsymbol{x}_i^{(t)} + \beta_t\boldsymbol{u}_i^{(t)}, \boldsymbol{x}_j^{(t)}) \right\}$$

$$=: \boldsymbol{x}_i^{(t)} - \gamma \cdot \mathscr{F}(\boldsymbol{x}_i^{(t)} + \beta_t\boldsymbol{u}_i^{(t)}),$$

where $\{\boldsymbol{x}_i^{(t)}\}_{i=1}^n$ are fixed and the only randomness in the update scheme above comes from the injected Gaussian noise $\{\boldsymbol{u}_i^{(t+1)}\}_{i=1}^n$. As a result, the probability in the statement of Lemma B.8 is over the distribution of $\{\boldsymbol{u}_i^{(t+1)}\}_{i=1}^n$. In the remainder of the proof, we omit the superscript and subscript $t+1$ for simplicity denoting $\boldsymbol{x}_i^{(t+1)}, \boldsymbol{u}_i^{(t+1)}, \beta_{t+1}$ as $\boldsymbol{x}_i, \boldsymbol{u}_i, \beta$.

Let $\xi_i\colon \mathbb{R}^d \to \mathcal{H}$ be the mapping

$$\boldsymbol{u}_i \mapsto k\left(\boldsymbol{x}_i - \gamma\mathscr{F}(\boldsymbol{x}_i + \beta\boldsymbol{u}_i), \ \cdot\right) - \mathbb{E}_{\boldsymbol{u}_i}\left[ k\left(\boldsymbol{x}_i - \gamma\mathscr{F}(\boldsymbol{x}_i + \beta\boldsymbol{u}_i), \ \cdot\right) \right].$$

for $i = 1, \ldots, n$. Since $\{\boldsymbol{u}_i\}_{i=1}^n$ are $n$ identical independent unit Gaussian random variables, $\{\xi_i(\boldsymbol{u}_i)\}_{i=1}^n$ are thus $n$ independent zero mean Hilbert-space valued random variables. For kernel $k$ that satisfies $\sup_{\boldsymbol{x}\in\mathbb{R}^d} k(\boldsymbol{x}, \boldsymbol{x}) \leq \kappa$ in Assumption 4, we know that $\sup_{\boldsymbol{u}} \|\xi_i(\boldsymbol{u})\|_{\mathcal{H}} \leq 2\sqrt{\kappa}$ for $i = 1, \ldots, n$, and

$$\mathbb{E}_{\boldsymbol{u}_i}\left[ \|\xi_i(\boldsymbol{u}_i)\|_{\mathcal{H}}^2 \right] = \mathbb{E}_{\boldsymbol{u}_i}\left[ \left\| k\left(\boldsymbol{x}_i - \gamma\mathscr{F}(\boldsymbol{x}_i + \beta\boldsymbol{u}_i), \ \cdot\right) - \mathbb{E}_{\boldsymbol{u}_i'}\left[ k\left(\boldsymbol{x}_i - \gamma\mathscr{F}(\boldsymbol{x}_i + \beta\boldsymbol{u}_i'), \ \cdot\right) \right] \right\|_{\mathcal{H}}^2 \right]$$

$$= \mathbb{E}_{\boldsymbol{u}_i}\left[ k\left(\boldsymbol{x}_i - \gamma\mathscr{F}(\boldsymbol{x}_i + \beta\boldsymbol{u}_i), \boldsymbol{x}_i - \gamma\mathscr{F}(\boldsymbol{x}_i + \beta\boldsymbol{u}_i)\right) \right]$$

$$- \mathbb{E}_{\boldsymbol{u}_i, \boldsymbol{u}'_i} \left[ k \left( \boldsymbol{x}_i - \gamma \mathscr{F}(\boldsymbol{x}_i + \beta \boldsymbol{u}_i), \boldsymbol{x}_i - \gamma \mathscr{F}(\boldsymbol{x}_i + \beta \boldsymbol{u}'_i) \right) \right]. \tag{22}$$

From Lemma B.11 we have,

$$\|\mathscr{F}(\boldsymbol{x}_i + \beta \boldsymbol{u}_i) - \mathscr{F}(\boldsymbol{x}_i + \beta \boldsymbol{u}'_i)\| = \left\| \frac{1}{n} \sum_{j=1}^n \Phi(\boldsymbol{x}_i + \beta \boldsymbol{u}_i, \boldsymbol{x}_j) - \frac{1}{n} \sum_{j=1}^n \Phi(\boldsymbol{x}_i + \beta \boldsymbol{u}'_i, \boldsymbol{x}_j) \right\|$$

$$\leq d\kappa\beta \|\boldsymbol{u}_i - \boldsymbol{u}'_i\| \operatorname{MMD}\left( \frac{1}{n} \sum_{j=1}^n \delta_{\boldsymbol{x}_j^{(t)}}, \mu \right) \leq \beta d \kappa^{3/2} \|\boldsymbol{u}_i - \boldsymbol{u}'_i\|.$$

Hence, we have

$$\mathbb{E}_{\boldsymbol{u}_i} \left[ \|\xi_i(\boldsymbol{u}_i)\|_{\mathcal{H}}^2 \right] \leq \mathbb{E}_{\boldsymbol{u}_i, \boldsymbol{u}'_i} \left[ \sqrt{d}\kappa\gamma \|\mathscr{F}(\boldsymbol{x}_i + \beta \boldsymbol{u}_i) - \mathscr{F}(\boldsymbol{x}_i + \beta \boldsymbol{u}'_i)\| \right]$$

$$\leq \gamma \beta d^{3/2} \kappa^{5/2} \, \mathbb{E}_{\boldsymbol{u}_i, \boldsymbol{u}'_i} [\|\boldsymbol{u}_i - \boldsymbol{u}'_i\|] =: \gamma \beta C_0.$$

$C_0$ is a positive universal constant. We have thus proved that $\sup_{\boldsymbol{u}} \|\xi_i(\boldsymbol{u})\|_{\mathcal{H}} \leq 2\sqrt{\kappa}$ and $\mathbb{E}_{\boldsymbol{u}_i}[\|\xi_i(\boldsymbol{u}_i)\|_{\mathcal{H}}^2] \leq \gamma \beta C_0$. Define $Z := \frac{1}{n} \sum_{i=1}^n \xi_i(\boldsymbol{u}_i)$. We apply the Bernstein's concentration inequality in Hilbert spaces [91, Theorem 6.14] to obtain, for any $\tau > 0$, with probability at least $1 - \exp(-\tau)$,

$$\|Z\|_{\mathcal{H}} \leq \sqrt{\frac{2C_0 \gamma \beta \tau}{n}} + \sqrt{\frac{C_0 \gamma \beta}{n}} + \frac{4\sqrt{\kappa}\tau}{3n}.$$

For $\tau > 1$, we have with probability at least $1 - \exp(-\tau)$,

$$\|Z\|_{\mathcal{H}} \leq C\tau n^{-\frac{1}{2}} ((\beta\gamma)^{\frac{1}{2}} \vee n^{-\frac{1}{2}}).$$

Here, $C$ is a positive universal constant independent of $n, \beta$. Finally, notice that

$$\|Z\|_{\mathcal{H}} = \left\| \frac{1}{n} \sum_{i=1}^n k(\boldsymbol{x}_i - \gamma \mathscr{F}(\boldsymbol{x}_i + \beta \boldsymbol{u}_i), \cdot) - \mathbb{E}_{\underline{\boldsymbol{u}}} \left[ \frac{1}{n} \sum_{i=1}^n k(\boldsymbol{x}_i - \gamma \mathscr{F}(\boldsymbol{x}_i + \beta \boldsymbol{u}_i), \cdot) \right] \right\|_{\mathcal{H}}$$

$$\geq \left\| \frac{1}{n} \sum_{i=1}^n k(\boldsymbol{x}_i - \gamma \mathscr{F}(\boldsymbol{x}_i + \beta \boldsymbol{u}_i), \cdot) - \mathbb{E}_{\boldsymbol{y} \sim \mu}[k(\boldsymbol{y}, \cdot)] \right\|_{\mathcal{H}}$$

$$- \left\| \mathbb{E}_{\underline{\boldsymbol{u}}} \left[ \frac{1}{n} \sum_{i=1}^n k(\boldsymbol{x}_i - \gamma \mathscr{F}(\boldsymbol{x}_i + \beta \boldsymbol{u}_i), \cdot) \right] - \mathbb{E}_{\boldsymbol{y} \sim \mu}[k(\boldsymbol{y}, \cdot)] \right\|_{\mathcal{H}}$$

$$\geq \left\| \frac{1}{n} \sum_{i=1}^n k(\boldsymbol{x}_i - \gamma \mathscr{F}(\boldsymbol{x}_i + \beta \boldsymbol{u}_i), \cdot) - \mathbb{E}_{\boldsymbol{y} \sim \mu}[k(\boldsymbol{y}, \cdot)] \right\|_{\mathcal{H}}$$

$$- \mathbb{E}_{\underline{\boldsymbol{u}}} \left[ \left\| \frac{1}{n} \sum_{i=1}^n k(\boldsymbol{x}_i - \gamma \mathscr{F}(\boldsymbol{x}_i + \beta \boldsymbol{u}_i), \cdot) - \mathbb{E}_{\boldsymbol{y} \sim \mu}[k(\boldsymbol{y}, \cdot)] \right\|_{\mathcal{H}} \right],$$

where the second line is triangular inequality and the third line is Jensen's inequality. The proof is concluded.

$\square$

### B.4.3. AUXILIARY RESULTS

**Lemma B.11.** *Under Assumption 1 and 4, the following inequalities hold:*

*(1)* $\|k(\boldsymbol{z}, \cdot) - k(\boldsymbol{z}', \cdot)\|_{\mathcal{H}} \leq \sqrt{d}\kappa \|\boldsymbol{z} - \boldsymbol{z}\|.$

*(2)* $\|\Phi(\boldsymbol{z}, \boldsymbol{w}) - \Phi(\boldsymbol{z}', \boldsymbol{w}')\| \leq 2d\kappa (\|\boldsymbol{z} - \boldsymbol{z}'\| + \|\boldsymbol{w} - \boldsymbol{w}'\|).$

*(3)* $\left\| \frac{1}{n} \sum_{j=1}^n \left( \Phi(\boldsymbol{z}, \boldsymbol{x}_j^{(t)}) - \Phi(\boldsymbol{z}', \boldsymbol{x}_j^{(t)}) \right) \right\| \leq d\kappa \|\boldsymbol{z} - \boldsymbol{z}'\| \operatorname{MMD}(\frac{1}{n} \sum_{j=1}^n \delta_{\boldsymbol{x}_j^{(t)}}, \mu).$

*Proof.* Notice that Assumption 2 of [46] is satisfied with $K_{1d} = d^2\kappa$ hence from Lemma 7 of [46] point (1) is proved. Next, notice that

$$
\begin{aligned}
\|\nabla_1 k(\boldsymbol{z}, \boldsymbol{w}) - \nabla_1 k(\boldsymbol{z}', \boldsymbol{w})\|^2 &= \sum_{i=1}^d (\partial_i k(\boldsymbol{z}, \boldsymbol{w}) - \partial_i k(\boldsymbol{z}', w))^2 \\
&\overset{(*)}{=} \sum_{i=1}^d \left( \nabla_u \partial_i k(\boldsymbol{u}, \boldsymbol{w})^\top (\boldsymbol{z} - \boldsymbol{z}') \right)^2 \\
&\leq \sum_{i=1}^d \|\nabla_u \partial_i k(\boldsymbol{u}, \boldsymbol{w})\|^2 \|\boldsymbol{z} - \boldsymbol{z}'\|^2 \\
&= \sum_{i=1}^d \left( \sum_{j=1}^d \left( \partial_i \partial_j k(\boldsymbol{u}, \boldsymbol{w}) \right)^2 \right) \|\boldsymbol{z} - \boldsymbol{z}'\|^2 \\
&\leq d^2 \kappa^2 \|\boldsymbol{z} - \boldsymbol{z}'\|^2.
\end{aligned}
$$

Here, since $\boldsymbol{z} \mapsto \partial_i k(\boldsymbol{z}, \boldsymbol{w})$ is continuous, $(*)$ holds by the mean value theorem that $|\partial_i k(\boldsymbol{z}, \boldsymbol{w}) - \partial_i k(\boldsymbol{z}', \boldsymbol{w})| = \nabla_u \partial_i k(\boldsymbol{u}, \boldsymbol{w})^\top (\boldsymbol{z} - \boldsymbol{z}')$ holds for some $\boldsymbol{u} = t\boldsymbol{z} + (1-t)\boldsymbol{z}'$ on the line segment between $\boldsymbol{z}, \boldsymbol{z}'$; and the last inequality holds by using Assumption 4. Similarly, we have $\|\nabla_1 k(\boldsymbol{z}, \boldsymbol{w}) - \nabla_1 k(\boldsymbol{z}, \boldsymbol{w}')\| \leq d\kappa \|\boldsymbol{w} - \boldsymbol{w}'\|$. Therefore,

$$
\begin{aligned}
&\|\Phi(\boldsymbol{z}, \boldsymbol{w}) - \Phi(\boldsymbol{z}', \boldsymbol{w})\| \quad &(23) \\
&= \left\| \left( \mathbb{E}_{\boldsymbol{y} \sim \mu}[\nabla_1 k(\boldsymbol{z}, \boldsymbol{y})] - \nabla_1 k(\boldsymbol{z}, \boldsymbol{w}) \right) - \left( \mathbb{E}_{\boldsymbol{y} \sim \mu}[\nabla_1 k(\boldsymbol{z}', \boldsymbol{y})] - \nabla_1 k(\boldsymbol{z}', \boldsymbol{w}) \right) \right\| \\
&\leq \|\mathbb{E}_{\boldsymbol{y} \sim \mu}[\nabla_1 k(\boldsymbol{z}, \boldsymbol{y}) - \nabla_1 k(\boldsymbol{z}', \boldsymbol{y})]\| + \|\nabla_1 k(\boldsymbol{z}, \boldsymbol{w}) - \nabla_1 k(\boldsymbol{z}', \boldsymbol{w})\| \\
&\leq 2d\kappa \|\boldsymbol{z} - \boldsymbol{z}'\|. \quad &(24)
\end{aligned}
$$

Similarly, we can prove that $\|\Phi(\boldsymbol{z}, \boldsymbol{w}') - \Phi(\boldsymbol{z}, \boldsymbol{w}')\| \leq 2d\kappa \|\boldsymbol{w} - \boldsymbol{w}'\|$. Combining it and (23) completes the proof of point (2).

Finally, notice that the Assumption 2 of [46] is satisfied with $K_{2d} = d^2\kappa$, hence from Lemma 7 of [46] we obtain

$$
\sum_{i=1}^d \|\partial_i k(\boldsymbol{z}, \cdot) - \partial_i k(\boldsymbol{z}', \cdot)\|_{\mathcal{H}}^2 \leq d^2 \kappa \|\boldsymbol{z} - \boldsymbol{z}'\|^2.
$$

Consequently, we obtain

$$
\begin{aligned}
&\left\| \frac{1}{n} \sum_{j=1}^n \left( \Phi(\boldsymbol{z}, \boldsymbol{x}_j^{(t)}) - \Phi(\boldsymbol{z}', \boldsymbol{x}_j^{(t)}) \right) \right\|^2 \\
&= \sum_{i=1}^d \left\langle \partial_i k(\boldsymbol{z}, \cdot) - \partial_i k(\boldsymbol{z}', \cdot), \mathbb{E}_{\boldsymbol{y} \sim \mu}[k(\boldsymbol{y}, \cdot)] - \frac{1}{n} \sum_{j=1}^n k(\boldsymbol{x}_j^{(t)}, \cdot) \right\rangle_{\mathcal{H}}^2 \\
&\leq \sum_{i=1}^d \|\partial_i k(\boldsymbol{z}, \cdot) - \partial_i k(\boldsymbol{z}', \cdot)\|_{\mathcal{H}}^2 \left\| \mathbb{E}_{\boldsymbol{y} \sim \mu}[k(\boldsymbol{y}, \cdot)] - \frac{1}{n} \sum_{j=1}^n k(\boldsymbol{x}_j^{(t)}, \cdot) \right\|_{\mathcal{H}}^2 \\
&\leq d^2 \kappa \|\boldsymbol{z} - \boldsymbol{z}'\|^2 \operatorname{MMD}^2 \left( \frac{1}{n} \sum_{j=1}^n \delta_{\boldsymbol{x}_j^{(t)}}, \mu \right),
\end{aligned}
$$

which concludes the proof of point (3). $\qquad\square$

**Lemma B.12** (Discrete Grönwall's lemma)**.** *If $a_{t+1} \leq (1 + \lambda_t)a_t + b_t$ for any $t = 1, \ldots,$ then $a_T \leq \prod_{t=1}^{T-1}(1 + \lambda_t)a_1 + \sum_{t=1}^{T-1} b_t \cdot \left( \prod_{s=t+1}^{T-1}(1 + \lambda_s) \right)$ for any $T \in \mathbb{N}^+$.*

*Proof.* The proof is trivial via induction. $\qquad\square$

# C. Additional Experimental Details

**Mixture of Gaussians:** The target distribution $\mu = \frac{1}{10}\sum_{m=1}^{10}\mathcal{N}(\boldsymbol{\mu}_m, \boldsymbol{\Sigma}_m)$ is a mixture of 10 two-dimensional ($d = 2$) Gaussian distributions following the set-up in [26, 57]. The kernel is taken to be Gaussian kernel $k(\boldsymbol{x}, \boldsymbol{y}) = \exp(-\frac{1}{2\ell^2}\|\boldsymbol{x} - \boldsymbol{y}\|^2)$.

First, we are going to provide the closed form expression for integration of the function $f : \boldsymbol{x} \mapsto \sum_{j=1}^{n}\sum_{\ell=1}^{d}\partial_\ell k(\boldsymbol{x}_j, \boldsymbol{x})$ against $\mu$. To this end, it suffices to consider the integration of $\nabla_1 k(\boldsymbol{x}_j, \boldsymbol{x})$ against a single Gaussian distribution $\mathcal{N}(\boldsymbol{\mu}_m, \boldsymbol{\Sigma}_m)$, denoted as $I_{j,m} \in \mathbb{R}^d$,

$$
\begin{aligned}
I_{j,m} &= \int \nabla_1 k(\boldsymbol{x}_j, \boldsymbol{x})\, \mathrm{d}\mathcal{N}(\boldsymbol{x}; \boldsymbol{\mu}_m, \boldsymbol{\Sigma}_m) \\
&= \int \exp\left(-\frac{1}{2}\ell^{-2}\|\boldsymbol{x}_j - \boldsymbol{x}\|^2\right) \cdot -\ell^{-2} \cdot (\boldsymbol{x}_j - \boldsymbol{x})\, \mathrm{d}\mathcal{N}(\boldsymbol{x}; \boldsymbol{\mu}_m, \boldsymbol{\Sigma}_m) \\
&= \det(\ell^{-2}\boldsymbol{\Sigma}_m + \mathrm{I})^{-\frac{1}{2}} \cdot \exp\left(-\frac{1}{2}\boldsymbol{\mu}_m^\top \boldsymbol{\Sigma}_m^{-1}\boldsymbol{\mu}_m\right) \\
&\quad \cdot \exp\left(-\frac{1}{2}\ell^{-2}\|\boldsymbol{x}_j\|^2 + \frac{1}{2}\left(2\ell^{-2}\boldsymbol{x}_j + \boldsymbol{\Sigma}_m^{-1}\boldsymbol{\mu}_m\right)^\top (\ell^{-2} + \boldsymbol{\Sigma}_m^{-1})^{-1}\left(\ell^{-2}\boldsymbol{x}_j + \boldsymbol{\Sigma}_m^{-1}\boldsymbol{\mu}_m\right)\right) \\
&\quad \cdot \left(-\ell^{-2}\boldsymbol{x}_j + \ell^{-2}(\ell^{-2} + \boldsymbol{\Sigma}_m^{-1})^{-1}\left(\ell^{-2}\boldsymbol{x}_j + \boldsymbol{\Sigma}_m^{-1}\boldsymbol{\mu}_m\right)\right).
\end{aligned}
$$

So the integral $\int f d\mu = \frac{1}{10}\sum_{m=1}^{10}\sum_{j=1}^{n}\mathbb{1}_d^\top I_{j,m}$.

| Dim $d$ | Method | $n = 100$ | $n = 300$ | $n = 1000$ |
|---|---|---|---|---|
| 10 | Stationary MMD | $0.000957 \pm 0.000029$ | $0.000711 \pm 0.000090$ | $0.000194 \pm 0.000008$ |
| 10 | IID | $0.001423 \pm 0.000364$ | $0.001195 \pm 0.000170$ | $0.000829 \pm 0.000145$ |
| 50 | Stationary MMD | $0.000937 \pm 0.000015$ | $0.000536 \pm 0.000008$ | $0.000138 \pm 0.000005$ |
| 50 | IID | $0.001585 \pm 0.000276$ | $0.000978 \pm 0.000172$ | $0.000620 \pm 0.000110$ |

**Table 1.** Comparison of our stationary MMD samples and i.i.d samples across dimensions $d$ and sample sizes $n$.

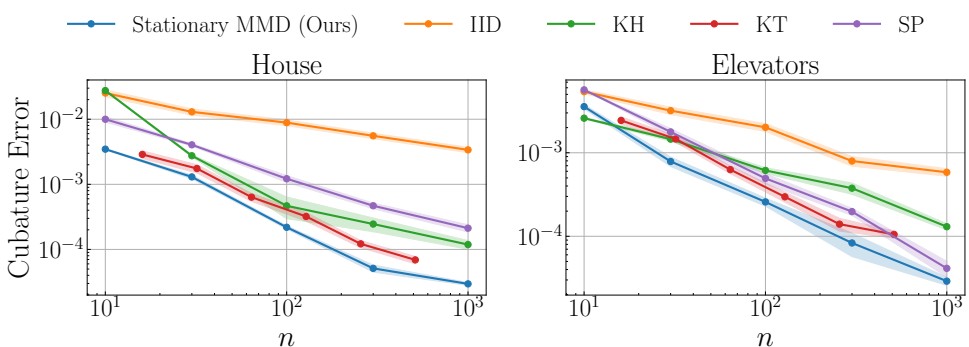

**Figure 5.** Ablation study of our stationary MMD points and all baselines using a Matérn-$\frac{3}{2}$ kernel on the *House8L* and *Elevators* datasets. The function used for integration is $f_1(\boldsymbol{x}) = \exp(-0.5\|\boldsymbol{x}\|^2)$.

Next, we are going to provide the closed form expression for integration of two functions $f_1 : \boldsymbol{x} \mapsto \exp(-0.5\|\boldsymbol{x}\|^2)$ and $f_2 : \boldsymbol{x} \mapsto \|\boldsymbol{x}\|^2$ against the distribution $\mu$, which are used as the ground truth integral value for benchmarking. Similarly, it suffices to consider the integration of $f_1, f_2$ against a single Gaussian distribution $\mathcal{N}(\boldsymbol{\mu}_m, \boldsymbol{\Sigma}_m)$.

$$
\int \exp\left(-\frac{1}{2}\|\boldsymbol{x}\|^2\right)\mathcal{N}(\boldsymbol{x}; \boldsymbol{\mu}_m, \boldsymbol{\Sigma}_m)\, \mathrm{d}\boldsymbol{x} = \exp\left(\frac{1}{2}\boldsymbol{\mu}_m^\top \boldsymbol{\Sigma}_m^{-1}\left(\boldsymbol{\Sigma}_m^{-1} + \mathrm{I}\right)^{-1}\boldsymbol{\Sigma}_m^{-1}\boldsymbol{\mu}_m\right)
$$
$$
\cdot \exp\left(-\frac{1}{2}\boldsymbol{\mu}_m^\top \boldsymbol{\Sigma}_m^{-1}\boldsymbol{\mu}_m\right) \cdot \left(\det\left(\boldsymbol{\Sigma}_m + \mathrm{I}\right)\right)^{-\frac{1}{2}},
$$
$$
\int \|\boldsymbol{x}\|^2 \mathcal{N}(\boldsymbol{x}; \boldsymbol{\mu}_m, \boldsymbol{\Sigma}_m)\, \mathrm{d}\boldsymbol{x} = \operatorname{tr}(\boldsymbol{\Sigma}_m) + \|\boldsymbol{\mu}_m\|^2.
$$

Now, we provide additional implementation details of our stationary MMD points and all baselines. We compute stationary MMD points by simulating the noisy MMD particle descent in Eq. (8). The number of iterations is set to $T = 10{,}000$ for $n = 10$, $T = 100{,}000$ for $n \in \{30, 100, 300\}$, and $n = 300{,}000$ for $n = 1000$, which we find sufficient for convergence. We adopt a fixed step size $\gamma = 1.0$ and do not inject additional noise, since the particles are initialized in a neighborhood of 0, which already yields stable and fast convergence. The step size is selected by searching over a small set of step sizes $\{0.1, 0.3, 1.0, 3.0\}$ and select the value that yields the best performance. We use a fixed kernel lengthscale $\ell = 1$ for our method and all baselines to ensure consistency and avoid introducing tuning bias. This choice prioritizes simplicity and reproducibility, and further performance gains are expected with more problem-specific lengthscale tuning. For the QMC baseline, we first assign an equal number of samples to each Gaussian component in the mixture. Then, we generate quasi-Monte Carlo (QMC) samples of each Gaussian component by transforming a Sobol sequence via the corresponding Gaussian inverse cumulative distribution function. We follow the technique introduced in randomized QMC [71] to shift the Sobol sequence by a random amount to account for the bias when the number of samples for each component are not a power of 2. For the kernel herding (KH) baseline, the greedy minimisation at each step is performed using L-BFGS-B from the `scipy.optimize` package, which eliminates the need to manually specify a learning rate or the number of iterations. For the support points (SP) baseline, we use the implementation in `https://github.com/kshedden/SupportPoints.jl`, as no official implementations are publicly available. In particular, we set the number of iterations to $T = 10{,}000$ for $n \in \{10, 30, 100\}$ and $T = 30{,}000$ for $n \in \{300, 1000\}$. For the kernel thinning (KT) baseline, we use the `kernel_split` and `kernel_swap` functions from `goodpoints.jax.kt` in the `goodpoints` library[8], instead of using `goodpoints.kt.thin` directly. This implementation allows customization of the kernel function and the oversampling parameter $\mathfrak{g}$ for our setting. `kernel_split` operates on a candidate set of size $n^2$, while `kernel_swap` operates on a candidate set of size $2^{\mathfrak{g}} n^2$; in both cases, the candidates are sampled i.i.d. from the mixture-of-Gaussians distribution. Other hyperparameters of KT include failure probability $\delta = 0.5$, `random_swap_order` set to True, `baseline` set to False, and number of KT-swap iterations `num_repeat` set to 100.

**OpenML Datasets:** We use *House8L* ($d = 8$) dataset and *Elevators* ($d = 18$) dataset from the OpenML database [16] where the target distribution $\mu$ is the discrete measure corresponding to the full dataset. Both dataset are normalized to have zero mean and unit standard deviation. The ground truth integral is the empirical average of the integrand $f_1$ over the whole dataset. As a result, we fix the kernel lengthscale to $\ell = 1$, rather than using the common heuristic of setting it to the median pairwise Euclidean distance. Our stationary MMD points are computed by simulating noisy MMD particle descent in Eq. (8). We fix the number of iterations to $T = 100{,}000$ for all values of $n$ and for both the *House8L* and *Elevators* datasets, which we find to be sufficient for convergence. We adopt a constant step size $\gamma = 1.0$ for the *House8L* datasets and $\gamma = 0.3$ for the *Elevators* dataset. The noise injection schedule is set to $\beta_t = \beta_0 t^{-1/2}$ with initial $\beta_0 = 1.0$. QMC is no longer applicable here since the domain of the datasets are unknown. The configurations for the KH, KT baselines are the same as in the mixture of Gaussian experiment. `kernel_split` operates on a candidate set of size $n_{\text{split}} = n^2$. When $n_{\text{split}}$ is smaller than $N$ the total size of the dataset, the set is sampled without replacement; when $n_{\text{split}} > N$, each of the $N$ points is replicated $\lfloor n_{\text{split}}/N \rfloor$ times, and the remaining points are sampled uniformly without replacement to reach a total of $n_{\text{split}}$ samples. `kernel_swap` operates the entire empirical dataset. For SP in both datasets, we set the number of iterations to $T = 10{,}000$ for $n \in \{10, 30, 100\}$ and $T = 30{,}000$ for $n \in \{300, 1000\}$.

### C.1. Ablation study with Matérn-$\frac{3}{2}$ kernel

In Figure 5, we present an ablation study of our stationary MMD points, KH (kernel herding) and KT (kernel thinning) using a Matérn-$\frac{3}{2}$ kernel of lengthscale $\ell = 1.0$. The Matérn-$\frac{3}{2}$ kernel satisfies all Assumption 1, 3 and 4 required for our stationary MMD points. SP (support points) remains the same because it does not involve choice of kernels [73]. The results are consistent with those obtained using Gaussian kernels in the main text. Notably, for our stationary MMD points, the Matérn-$\frac{3}{2}$ kernel even outperforms the Gaussian kernel on the *House8L* dataset. A more comprehensive investigation of kernel choices is left for future work.

### C.2. More Experiments on Kernel Thinning

When the first version of this work appeared on arXiv [29], we were contacted by Lester Mackey, who kindly pointed out that our implementation KT was potentially unfair. In particular, the implementation of KT in [29] does not exploit closed-form expressions for kernel mean embeddings when computing the MMD, but instead estimates the MMD using samples. In

---

[8]`https://github.com/microsoft/goodpoints`

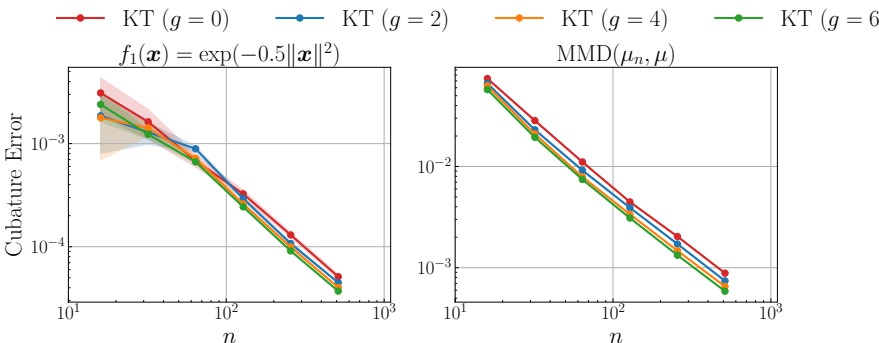

**Figure 6.** Comparison of KT in the mixture of Gaussian setting with different values of the oversampling parameter $\mathfrak{g}$.

contrast, both kernel herding and stationary MMD point methods make use of closed-form kernel mean embeddings. As a result, he suggested that employing a *centered* kernel as follows can substantially improve the performance of KT baselines. For a Gaussian kernel $k$, the associated *centered* Gaussian kernel $\tilde{k}$ is

$$\tilde{k}(\boldsymbol{x}, \boldsymbol{y}) = k(\boldsymbol{x}, \boldsymbol{y}) - \mathbb{E}_{\boldsymbol{Y} \sim \mu}[k(\boldsymbol{x}, \boldsymbol{Y})] - \mathbb{E}_{\boldsymbol{Y} \sim \mu}[k(\boldsymbol{Y}, \boldsymbol{y})] + \mathbb{E}_{\boldsymbol{Y} \sim \mu, \boldsymbol{Y}' \sim \mu}[k(\boldsymbol{Y}, \boldsymbol{Y}')].$$

For the mixture of gaussian setting, both $\mathbb{E}_{\boldsymbol{Y} \sim \mu}[k(\boldsymbol{x}, \boldsymbol{Y})]$ and $\mathbb{E}_{\boldsymbol{Y} \sim \mu, \boldsymbol{Y}' \sim \mu}[k(\boldsymbol{Y}, \boldsymbol{Y}')]$ admit closed-form expressions. For the *House8L* and *Elevators* setting, since $\mu$ is an empirical distribution, both terms reduce to empirical averages. After centering, the integral of the kernel under $\mu$ is zero, i.e., $\mathbb{E}_{\boldsymbol{x} \sim \mu}[\tilde{k}(\boldsymbol{x}, \boldsymbol{y})] = 0$.

Next, in Figure 7, we compare the performance of KT when using an *uncentered* Gaussian kernel $k$ versus a *centered* Gaussian kernel $\tilde{k}$. The uncentered kernel corresponds to the choice adopted in the first version of this paper [29], which we refer to as KT (old) in Figure 7. Specifically for KT (old), we directly use `goodpoints.kt.thin`, with `split_kernel` and `swap_kernel` both set to the uncentered Gaussian kernel. All remaining hyperparameters are left at their default values. We confirm that using a *centered* kernel indeed greatly improves the performance of KT.

Additionally, we also compare the performance of KT and in the mixture of Gaussian setting with different values of the oversampling hyperparameter $\mathfrak{g} \in \{0, 2, 4, 6\}$. As shown in Figure 6, the performance of both methods consistently improves as $\mathfrak{g}$ increases. However, using a $\mathfrak{g} > 0$ requires access to a pre-specified set of samples of size $2^{\mathfrak{g}} N$ from $\mu$, which may not be available in practice. Also, a larger $\mathfrak{g}$ results in much higher computational cost in practice.

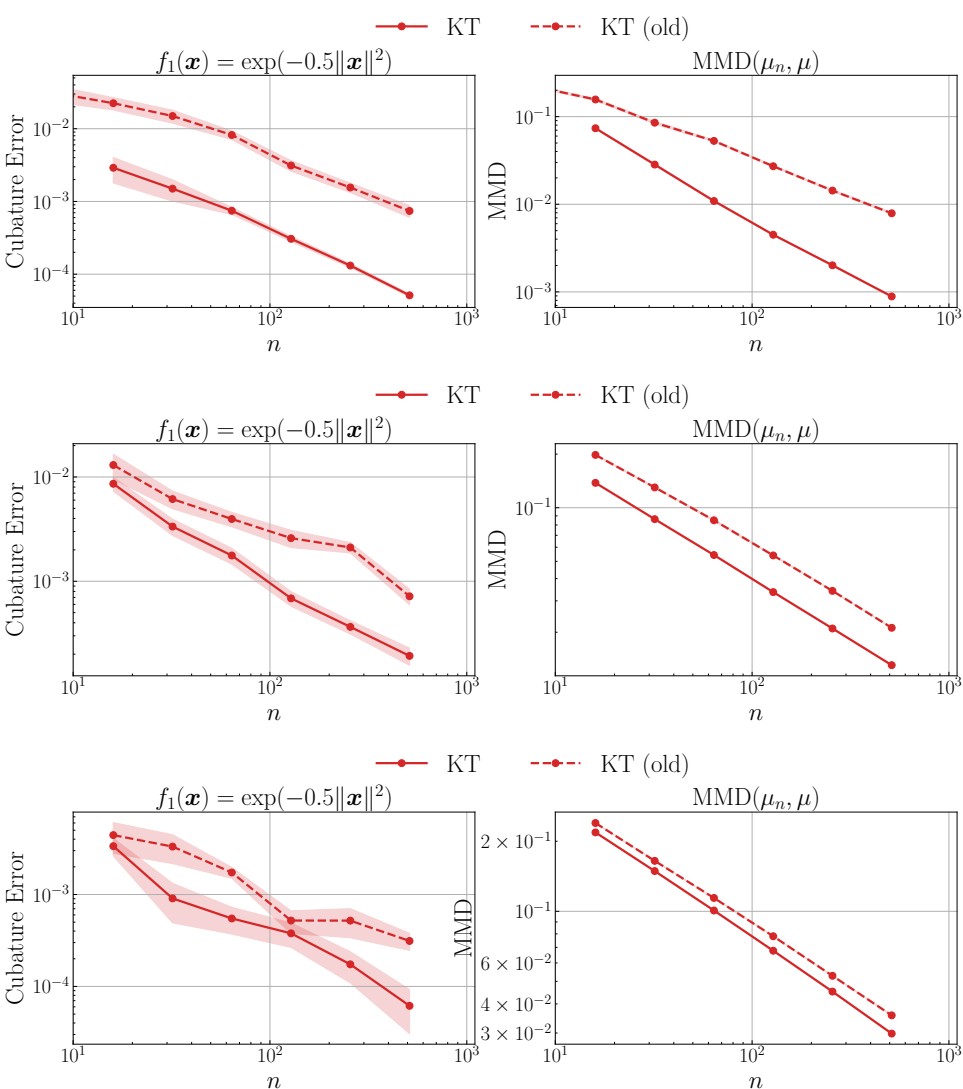

**Figure 7.** Comparison of the baselines KT across different datasets using *centered* and *uncentered* Gaussian kernels. **Top:** Mixture of Gaussians. **Middle:** *House8L* dataset. **Bottom:** *Elevators* dataset.

