# OpenReview forum: "Stationary MMD Points"
_ICML.cc/2026/Conference — ICML 2026 regular_

### Official Review · Reviewer_Aat4 · 2026-03-12

**Soundness:** 3
**Presentation:** 3
**Significance:** 3
**Originality:** 3
**Overall Recommendation:** 5
**Confidence:** 3

**Summary:**

This paper studies *stationary MMD points* rather than globally optimal minimum-MMD point sets. The motivation is compelling: while minimizing MMD globally is non-convex and generally not tractable in practice, stationary points can be reached by gradient-based procedures and are therefore a more realistic object of analysis.

The first main contribution is a theoretical result showing that stationary MMD points exactly integrate a nontrivial linear subspace of the RKHS, and that this exactness leads to a *super-convergence* result: for a broad class of integrands, the numerical integration error decays faster than the MMD itself. This helps explain why practically computed local minima or stationary configurations may still perform very well for integration.

The second main contribution is algorithmic. The authors analyze a noisy MMD particle descent scheme and prove a non-asymptotic finite-particle error bound. Under their assumptions, this leads to convergence toward stationary MMD points in the joint large-particle / long-time limit.

**Compliance With Llm Reviewing Policy:**

Affirmed.

**Key Questions For Authors:**

## Minor Comments

1. The exposition around the exactness space is mathematically elegant but could be made much more intuitive. I would suggest adding a short paragraph in the main text explaining \(G_{X_n}\) as the span of "local directional kernel features" and explicitly linking stationarity to quadrature exactness.

2. The paper should explicitly state that stationary MMD points are generally not unique. Even a short remark would help avoid confusion.

3. The practical discussion of computing
   \[
   \mathbb{E}_{Y\sim\mu}[\nabla_1 k(\cdot,Y)]
   \]
   should be expanded.

4. The experiments are encouraging, but the connection between theory and experiment could be sharpened. In particular, some experiments are conducted in settings where the support assumptions used in the super-convergence theory do not hold exactly. The paper acknowledges this, but a more explicit discussion of what is expected to remain valid would strengthen the presentation.

5. Initialization seems important in practice, especially given Remark 4.5. The empirical section would benefit from a brief discussion of how sensitive the method is to initialization and whether the observed gains are robust across different starts.

**Limitations:**

yes

**Strengths And Weaknesses:**

## Major Comments

1. **Uniqueness of the stationary MMD points.**

   The paper does not appear to establish uniqueness of stationary MMD points, and I do not think uniqueness should be expected in general.

   First, at a basic level, even if a configuration were unique geometrically, it would still only be unique up to permutation of the particle labels. More importantly, because the MMD objective is non-convex, multiple stationary points and multiple local minima are entirely plausible, especially for multimodal or symmetric target distributions.

   Indeed, the framing of the paper itself suggests non-uniqueness is natural: the whole motivation is that practical algorithms can get trapped in local minima, and stationary points form a strict relaxation of minimum-MMD points. That perspective is much more consistent with multiplicity than with uniqueness.

   For this reason, I would ask the authors to explicitly clarify the status of uniqueness of the stationary MMD points.

2. **How is the expectation in Eq. (8) computed? Are the final results exact-oracle results? What about inexact steps?**

   The update in Eq. (8) requires the term
   \[
   \mathbb{E}_{Y\sim\mu}[\nabla_1 k(z,Y)].
   \]
   I think the discussion should be expanded substantially.

   As I understand it, the theoretical algorithm and the final convergence results are based on *exact computation* of this expectation. The authors explain that this is feasible in several special cases:
   * for some kernels and target distributions, closed-form kernel mean embeddings are available,
   * change-of-variable tricks can sometimes make the expectation tractable,
   * Stein reproducing kernels can be used when the target is only known up to normalization.

   So my reading is that the stated theorems are indeed **exact-oracle results**. The only stochasticity explicitly analyzed in the paper comes from the injected Gaussian noise and the finite-particle concentration arguments, not from approximating the expectation over \(\mu\). This should be stated more explicitly.

   What can be said in the inexact case? At a high level if the expectation is replaced by an *unbiased estimator* with controlled variance, one might hope to obtain a perturbed version of the analysis, with an additional variance term in the finite-time bound.

3. **Comments on Assumption 5 and practical impact.**

   Assumption 5 is central to the convergence analysis. The authors explain that Assumption 5 can be checked a posteriori by sampling many realizations of the injected noise and selecting a noise level from a candidate set. This is a reasonable proof-guided heuristic, but it leaves several practical questions open:
   * How expensive is this empirical verification?
   * How sensitive is performance to the candidate grid for \(\beta_t\)?
   * What happens if none of the candidate values appears to satisfy the condition?

---

> ### Author Rebuttal · Authors · 2026-03-28
>
> ## General comment
> We sincerely thank the reviewer for the careful and supportive assessment of our work. We are especially encouraged that the reviewer found the motivation “compelling,” viewed stationary points as “a more realistic object of analysis,” and noted that the super-convergence result “helps explain why practically computed local minima or stationary configurations may still perform very well for integration.” We also greatly appreciate the reviewer’s positive assessment of our algorithmic contribution and the overall significance of the paper.
>
> ## Major 1: Uniqueness of the stationary MMD points.
> We appreciate the reviewer’s sharp observation. We agree that stationary MMD points should not be expected to be unique, even modulo permutation of the particle labels. For instance, in the four-cross example shown in Figure 1, it is highly plausible that the configuration obtained in the fourth panel is not unique: the six particles currently located on one cross could instead move to another cross and yield a different stationary configuration with similar qualitative behavior. Importantly, neither our algorithm nor our theory relies on uniqueness; the method remains well defined and the theory applies to any configuration computed. We will clarify this point in the camera-ready version.
>
> ## Major 2: How is the expectation in Eq. (8) computed? Are the final results exact-oracle results? What about inexact steps?
> We thank the reviewer for raising this point. The expectation in Eq. (8) is assumed to be available in closed form; see the paragraph below Eq.(8) (lines 254--264) for a detailed discussion, which we will further clarify in the revision. In all experiments considered in the paper, this expectation can indeed be computed in closed form, so the reported results correspond to an exact-oracle setting. If one instead replaces this expectation by an unbiased estimator with controlled variance, as the reviewer suggests, then an additional variance term would appear in the upper bound of Theorem 4.1. We are happy to add a brief remark after Theorem 4.1 to make this extension explicit.
>
> ## Major 3: Comments on Assumption 5 and practical impact.
> 1.How expensive is this empirical verification?
>
> We thank the reviewer for this practical question. Suppose the cost of evaluating the gradient at a single iteration is $C$. Then empirically verifying Assumption 5 with $M$ Gaussian perturbations requires averaging over these $M$ noise realizations, leading to a cost on the order of $MC$. Thus, verification is indeed more expensive than a single step of the original flow simulation. That said, the computation is highly parallelizable across the noise samples, so we expect the practical overhead to be reduced with vectorized implementations on GPUs.
>
> 2. How sensitive is performance to the candidate grid for $\beta\_t$?
>
> To assess sensitivity to the choice of $\beta_t$, we reran the mixture-of-Gaussians experiments with several alternative schedules; see the additional results here: https://drive.google.com/file/d/1sbz-1QhR09nx31zQ-pKayDVQTHkG4Ofp/view?usp=sharing. Specifically, we considered $\beta_t \propto t^{-1.0}$, $\beta_t \propto t^{-0.75}$, $\beta_t \propto t^{-0.5}$, $\beta_t \propto t^{-0.25}$, and $\beta_t = 0$. We find that, in this simple setting, the performance of stationary MMD points is broadly stable across these choices. That said, when the number of particles is relatively large, e.g. $n=1000$, different noise injection schemes do lead to noticeable differences. This indicates that the choice of noise schedule is more crucial in the large-$n$ regime. We will add this sensitivity analysis to the paper and explicitly note this limitation in the revision.
>
> 3. What happens if none of the candidate values appears to satisfy the condition?
>
> If none of the candidate values satisfies the condition, a natural practical remedy is first to enlarge or refine the candidate set for $\beta_t$. If the condition still fails for all such choices, then this indicates our convergence guarantee is therefore not directly applicable in such a setting. We view this as a limitation of the present theory rather than of the algorithm itself, and we will clarify this point in the revised paper.
>
>
> ## Minor comments:
> We thank the reviewer for these helpful suggestions.
>
> 1. We will add a short paragraph in the main text interpreting $G_{X_n}$ as the span of local directional kernel features and explicitly explaining how stationarity is connected to quadrature exactness.
>
> 2. See response to Major comments 1.
>
> 3. See response to Major comments 2.
>
> 4. Although the support condition does not hold in some of our experiments, we still expect stationarity to improve the empirical performance of numerical integration. We will add a short discussion to make this point clear in the revised paper.
>
> 5. We will add a brief discussion in the empirical section on sensitivity to the initialization in the revised version.

---

> > ### Author Rebuttal · Reviewer_Aat4 · 2026-04-01
> >
> > I thank the authors for their detailed response. I am increasing my score to accept.

---

### Official Review · Reviewer_9xC5 · 2026-03-13

**Soundness:** 4
**Presentation:** 3
**Significance:** 3
**Originality:** 3
**Overall Recommendation:** 5
**Confidence:** 3

**Summary:**

This paper studied stationary MMD points, where, similar to minimal MMD points, it exhibits strong and surprising numerical integration properties and are more amenable to being found computationally.

**Compliance With Llm Reviewing Policy:**

Affirmed.

**Final Justification:**

My main concern is the practical usefulness of the stationary MMD. The authors have provided some information in the rebuttal.

**Key Questions For Authors:**

1. In what situations are stationary MMD points more useful or easier to obtain than minimal MMD points? Can the authors provide some examples?

2. In what situations are minimal MMD points still preferred, and why?

3. How practically useful is the super convergence result, especially given this function class limitation?

**Limitations:**

1. The main theatrical result (super convergence in section 3) only holds for a specific function class.

**Strengths And Weaknesses:**

Strength:

1. This paper proposed a novel and nontrivial theoretical idea for MMD. MMD itself is also a very useful tool in many areas of ML (i.e., optimal transport and reinforcement learning), so the advancements are much appreciated.

2. The paper is generally well-written and easy to read. The main idea and contibution is presented clearly with a comprehensive discussion of related works. Connections between minimal energy points and kernel herding are nice to have.

3. Theoretical results are quite surprising, where the author showed that stationary MMD points, despite not being optimal, still have strong numerical integration properties.

Weakness:
1. Not a huge issue, but some additional discussion with regard to some actual applications of MMD or its minimal points would give reader more context of the paper and its contribution.

2. The core takeaway of this papers seems to be that stationary MMD points are a more computationally feasible to find than minimum-MMD points (and those are often just local minimal). However, the paper does not sufficiently discuss when this minimal to stationary relaxation is preferable or when one should still choose minimum-MMD points over stationary MMD points or vice versa.

3.  The main theorem does not say that stationary points are optimal (in any sense) nor says if they beat minimum-MMD points in any fashion. The usefulness of stationary MMD points needs to be further examined.

4. The super convergence result is limiting in terms of function classes (f = c + h). This is mathematically interesting, but the practical usefulness of the function class is unclear.

---

> ### Author Rebuttal · Authors · 2026-03-28
>
> ## General comment
> We sincerely thank the reviewer for the careful and supportive assessment of our work. We are especially encouraged that the reviewer viewed the paper as “proposing a novel and nontrivial theoretical idea for MMD”, found the “theoretical results surprising and interesting”, and felt that “the paper is generally well written and easy to read”. We also greatly appreciate the reviewer’s positive assessment that the paper is technically solid and advances an important line of work that others are likely to build on.
>
> ## W1: Some additional discussion with regard to some actual applications of MMD or its minimal points would give the reader more context of the paper and its contribution.
> We thank the reviewer for this helpful suggestion. We agree that adding more application-level context would improve the presentation. In the setting of our paper, minimum MMD points provide the best $n$-point approximation of a target distribution $\mu$ under the MMD criterion. Beyond the theoretical interest, MMD and minimum-MMD point sets have already found use in several practical areas, including sampling [28, 88, https://arxiv.org/pdf/2105.09994], dataset compression / coreset selection, which would reduce the computational cost of downstream tasks [24, 66], and numerical integration [32,35,36]. We will add a short discussion in the related work section to better highlight these applications and thereby help the readers to better understand the contribution of the paper in a broader practical context.
>
> ## W2 &W3 & Q2: When are minimum MMD points preferred over stationary MMD points? The main theorem does not say that stationary points are optimal (in any sense) nor says if they beat minimum-MMD points in any fashion.
> We thank the reviewer for this important question. We believe the key issue being raised here is whether stationary MMD points could be substantially worse than minimum MMD points. At present, we do not have a satisfactory general answer to this question, and we will make this limitation explicit in the revision.
> In principle, minimum MMD points are optimal for the MMD objective, since they achieve the smallest MMD over all $n$-particle configurations. However, computing them requires solving a highly challenging **non-convex global optimization problem**, and to the best of our knowledge, no existing work has established a general procedure for identifying minimum MMD points. There do exist approximate methods, which we discuss in the related work section 2. By contrast, stationary MMD points offer a more computationally tractable alternative, while still enjoying strong compression and numerical integration performance, as we have shown both theoretically in Theorem 3.4 and empirically in Section 5.
>
> We also note that our super-convergence result applies not only to stationary MMD points, but also to minimum MMD points, **since every minimum MMD point is necessarily stationary**. Thus, from the perspective of our theory, minimum MMD points are covered as a special case, even though their computation appears to be much more challenging.
>
> ## Q1: In what situations are stationary MMD points more useful or easier to obtain than minimal MMD points? Can the authors provide some examples?
> We thank the reviewer for this important question. In most practical settings, stationary MMD points are substantially easier to obtain than minimum-MMD points, because they can be exactly computed by our proposed noisy gradient descent algorithm, whereas minimum-MMD points require solving a highly non-convex global optimization problem. A simple illustration is given by the four-cross example in Figure 1: the stationary MMD points can be computed explicitly and correspond to a local minimum of the MMD objective, whereas the minimum-MMD points would presumably allocate the particles more evenly across the four crosses but cannot be computed in general.
>
> ## Q3 & W4: How practically useful is the super convergence result, especially given this function class limitation?
> We thank the reviewer for this important question. First, we note that no single cubature rule can be expected to perform well uniformly over all possible integrands; at best, one can hope for strong guarantees over a sufficiently rich and practically relevant function class. In our case, this class is the reproducing kernel Hilbert space (RKHS) associated with the kernel.
> When the kernel is $c_0$​-universal, the corresponding RKHS is **dense in $C_0(\mathcal{X})$**, the space of continuous functions vanishing at infinity [83]. This includes commonly used kernels such as the Gaussian and Matérn kernels considered in our paper. Therefore, although our super-convergence result is formally stated for functions in the RKHS, this class is already quite rich and can approximate a broad range of functions of practical interest. We will make this clear in the camera-ready version of this paper.

---

> > ### Author Rebuttal · Reviewer_9xC5 · 2026-04-02
> >
> > Thank you for the clarification; my questions are mostly answered to my satisfaction. I will adjust my score accordingly. I also encourage the authors to update the manuscript to reflect the practical and application side contribution of stationary MMD points.

---

### Official Review · Reviewer_725L · 2026-03-13

**Soundness:** 3
**Presentation:** 3
**Significance:** 4
**Originality:** 4
**Overall Recommendation:** 5
**Confidence:** 3

**Summary:**

This paper shows that $n$-particle stationary points of the MMD (maximum mean discrepancy) to a fixed probability distribution $\mu$---stationary in the Wasserstein sense---can be used for cubature with theoretical guarantees. Previously, this was only known for $n$-particle minimizers of the MMD, which are difficult to compute, while stationary points of the MMD are relatively easy to compute. The key insight of this work is that if $\mu_n = \frac1n \sum_i \delta_{x_i}$ is stationary for the MMD, then cubature is exact for all functions in $\mathcal{F}\_n$, the linear span of $\{k(x_i, \cdot)\}\_{i \leq n}$, and this space grows to cover all functions in the RKHS if $\mu\_n$ converges to $\mu$ in MMD. The paper also presents an algorithm for computing such $\mu_n$ with a convergence guarantee.

**Compliance With Llm Reviewing Policy:**

Affirmed.

**Final Justification:**

I maintain my recommendation to accept this paper, purely on the basis of the impressive contributions of its sections 1-3. I have not checked carefully the claims of section 4, though nothing particularly stood out.

**Key Questions For Authors:**

0. Correct me if any of the specific points I wrote in "Strengths and Weaknesses" are wrong.
1. Can the convergence in Proposition 3.3 be made quantitative with more knowledge about the kernel (and maybe the regularity of $f$)?
2. line 288 second column: "We highlight a trade-off between the optimisation and estimation errors: [...]" --- and how does the tradeoff look in terms of how likely we can expect Assumption 5 to be verified?
3. There is no universal constant fudge factor on the right-hand side of Assumption 5. Is this simply for the sake of presentation, or is it really important that the constant be $1$?
4. Empirically, is Assumption 5 ever not satisfied?
5. line 312: "In contrast, our assumption only needs to be checked for the current trajectory." This is up to the presence of the (admittedly expectedly fast concentrating) expectation w.r.t. the $u_i$, right? Or is it the case that if it is satisfied for a given random run, then guarantees follows for the output of that particular run?
6. line 339, second column: you say that in this experiment, "Assumption 5 is empirically satisfied" --- but for which n?

**Limitations:**

Yes

**Strengths And Weaknesses:**

The paper makes a valuable contribution to the fields of cubature and particle methods by introducing a significant new idea: that in the convergence of $\int f d\mu_n$ to $\int f d\mu$ for $\mu_n$ obtained through MMD Wasserstein gradient flow, the condition $MMD(\mu_n, \mu) \to 0$ comes into play not only via the rough upper bound $\int f d(\mu_n-\mu) \leq \\|f\\| MMD(\mu_n, \mu)$, but also through the growth of $\mathcal{F}_n$ to all of the RKHS (or the subset thereof which is relevant to integration against $\mu$). It is plausible that this insight will lead to further developments in the direction of particle methods with kernels.

The presentation of the paper is quite good overall. There is a potential issue in the fact that the term "stationary MMD points" includes a convergence condition. The authors were careful not to let this overloading of term affect the clarity of the presentation of the present paper; but it might lead to some confusion further down the road.
A few specific points (I don't list them as key questions since they're not questions):
- line 84, "they can be obtained via gradient-based optimisation simulated to a local minimum" can be a bit misleading for a reader who is new to the field, as it can be read as "*any* of the sensible gradient methods that have been proposed for MMD works"---which, to my understanding, is not known to be the case
- line 176, second column, "and subsets with non-empty interiors"---but here $\mathcal{X}$ is typically an open set in the first place if, say, $\mu$ is given as the Gibbs measure of a potential; and I don't know of any setting where it wouldn't be one
- line 280 Assumption 5: it would be helpful, for the first reading, to indicate whether one should think of $\beta_t$ as fast decaying, slow decaying, or if there is a tradeoff
- line 416, the section in the appendix where Table 1 can be found should be specified, for the sake of readers that cannot use the hyperlinks

I have not checked any of the proofs, so my rating for "Soundness" should be taken with precaution.

---

> ### Author Rebuttal · Authors · 2026-03-28
>
> ## General comments
> We thank the reviewer for the careful and thoughtful reading of our paper. All of the comments are insightful, directly to the point, and will lead  to substantial improvements in the paper. We are also very happy to hear the reviewer’s assessment that ``the paper makes a valuable contribution to the fields of cubature and particle methods.''
> ## Q0: Correct me if any of the specific points I wrote in "Strengths and Weaknesses" are wrong.
> Thank you for all the comments - none of the points were wrong and we answer more precisely some of these below:
>
> Line 84: This is a very good point, and we will revise the sentence to read: ``they can be obtained via the specific noisy gradient descent scheme proposed in Section 4".
>
> Line 176: You are right that in most applications $\mathcal{X}$ is an open set and therefore already has non-empty interior; our intention here was to exclude degenerate cases such as supports consisting only of Dirac masses.
>
> Line 280: $\beta_t$ should be thought of as slowly decaying, with a specific trade-off (see our response to your Q2).
>
> Line 416: Thank you; we will specify the exact appendix section containing Table 1.
>
> ## Q1: Can the convergence in Proposition 3.3 be made quantitative
> Thank you for this very insightful question. We thought carefully about this issue while writing the paper. The main technical difficulty is to control the RKHS norm of the residual, $\\|r\_n\\|\_{\\mathcal{H}}$. If only a supremum norm bound were needed $\\|r\_n\\|\_{\infty}$, then standard results such as in Wendland's Scattered data approximation book [Theorem 11.11] would likely yield a non-asymptotic rate, after some mild adaptations. However, for our later analysis in Theorem 3.4, we require a bound on $\\|r\_n\\|\_{\\mathcal{H}}$, and the RKHS norm is substantially stronger than the supremum norm. For this reason, we are only able to obtain asymptotic convergence. We would be very interested in this question further, as obtaining a sharp quantitative bound appears to require new ideas.
> ## Q2: How does the tradeoff look in terms of how likely we can expect Assumption 5 to be verified?
> Thank you for pointing this out. This is again a very sharp and insightful question. Both the left- and right-hand sides of Assumption 5 depend on the noise level $\beta_t$. A large noise level $\beta_t$ has a benign regularizing effect on the trajectory: this is reflected in the left-hand side of Assumption 5, since the Gaussian perturbation helps avoid poor local behavior. However, if $\beta_t$ is too large, then the right-hand side of Assumption 5 also increases, so the condition may cease to hold. Therefore, the noise level must be chosen large enough for the regularization effect to take place, which corresponds to the convergence term (the first term in Theorem 4.1), but not so large that the variance dominates (the second term in Theorem 4.1). We will add a discussion addressing this point to the camera-ready version.
> ## Q3: There is no universal constant fudge factor on the right-hand side of Assumption 5.
> The exact value of the constant is not essential. Our analysis would still go through if the right-hand side of Assumption 5 were multiplied by any universal positive constant; this would only modify the numerical constants in the downstream bounds in Theorem 4.1. We stated the assumption with constant 1 (equivalently, with the factor 4 as written) mainly to keep the presentation clean.
> ## Q4 : Empirically, is Assumption 5 ever not satisfied?
> In our experiments, we did not observe violations of Assumption 5 along the trajectories we tested. Empirically, it appears to be a mild condition at least in the regimes considered in the paper. That said, we do not want to overstate this point: we have not established that it would hold universally, and its validity would depend on the choice of $\beta_t$.
> ## Q5: line 312: This is up to the presence of the expectation right? Or is it the case that if it is satisfied for a given random run, then guarantees follow for the output of that particular run?
> For the first question: yes, exactly. For the second question: the intent of this sentence is mainly to contrast our assumption with closely-related noise-injection conditions in [3, 42], where one needs to control an additional expectation over the trajectory. In our case, the assumption is formulated only in terms of the current realized trajectory, rather than an extra average over all possible trajectories, which makes our assumption simpler to state and check. We will clarify this point for the camera-ready version of the paper.
> ## Q6: line 339, second column: you say that in this experiment, "Assumption 5 is empirically satisfied" --- but for which n?
> We apologize for this oversight. We empirically verify Assumption 5 in our experiments with a choice of $n=100$ in Figure 4. Similar conclusions are observed for other choices of $n$. We will make this clear in the camera-ready version.

---

> > ### Author Rebuttal · Reviewer_725L · 2026-04-01
> >
> > My Q5 was not formulated clearly, apologies. My concern is that Assumption 5 involves an expectation over $u^{(t)}_i$, but $u^{(t)}_i$ appears in the definition of the algorithm (Equation (8)). So it looks like Assumption 5 is about more than just the current trajectory, since it involves an expectation over realizations of the trajectory. Could you please clarify this point?

---

> > > ### Author Response · Authors · 2026-04-01
> > >
> > > We appreciate the reviewer’s continued engagement with our paper. Given a realized trajectory, Assumption 5 can be checked _a posteriori_: for each $t \\in \\{0,\ldots,T\\}$, one may draw many samples of $\boldsymbol{u}_i^{(t)}$ and empirically verify whether the condition holds at each iterate. This is what we mean that "our assumption only needs to be checked for the current trajectory" in line 311-312 of the paper.
> > >
> > > By contrast, we understand the reviewer’s point as concerning _a priori_ verification, that is, checking the condition before a trajectory is realized. In that case, we agree that such a verification would involve expectations over possible realizations of the trajectory, since the next iterate would be dependent on the injected noise $\boldsymbol{u}_i^{(t)}$. We thank the reviewer for pressing on this point, as it helped us identify that our presentation in the paper was not sufficiently clear. We will revise the paper to make this distinction explicit. Thank you again for this helpful comment.

---

### Official Review · Reviewer_w7DC · 2026-03-19

**Soundness:** 4
**Presentation:** 3
**Significance:** 3
**Originality:** 3
**Overall Recommendation:** 5
**Confidence:** 4

**Summary:**

I like the main observation in this paper. The authors study stationary points of the equal-weight MMD objective and show that such point sets exactly integrate a finite-dimensional space $F_n = \mathrm{span}\bigl(\{1\} \cup G_{X_n}\bigr)$, where \(G_{X_n}\) is spanned by kernel derivatives at the selected points. This is Proposition 3.1, and it follows directly from the stationarity condition in equation 4. The main theorem then combines this exactness with an approximation argument to obtain a super-convergence bound for numerical integration relative to the MMD error. The paper also proposes a noisy MMD particle descent scheme to compute such stationary point sets.

**Compliance With Llm Reviewing Policy:**

Affirmed.

**Key Questions For Authors:**

what exactly is the conceptual role of injected noise in Assumption 5? Is the claim that Gaussian perturbations regularize the particle objective sufficiently to obtain a PL/\L{}ojasiewicz inequality in expectation along the trajectory?

**Limitations:**

yes

**Strengths And Weaknesses:**

Strenght:
Proposition 3.1 is simple but elegant: once one writes down the stationarity condition, it becomes clear that the empirical measure is exact on the span of the kernel derivative features attached to the support points. This gives a clean quadrature interpretation of stationary MMD points, and to me this is the conceptual heart of the paper. I would also promote Lemma B.4 to the main body of the paper

Theorem 3.4 is also interesting. It takes $f = f_n + r_n$, and shows that the approximation error \(|r|_n\) goes to zero asymptotically for the relevant class of functions.

Weakness:
I would appricate more intuition behind Proposition 3.3

The paper states that, with a suitable choice of noise level \(\beta_t\), the noisy particle system satisfies a gradient-dominance / Polyak--\L{}ojasiewicz-type condition in expectation. But this is exactly the kind of property one would want to prove rather than assume.

---

> ### Author Rebuttal · Authors · 2026-03-28
>
> ### General comments
> We sincerely thank the reviewer for the careful reading of our manuscript and for the very positive and insightful assessment of our work. We are especially encouraged by the reviewer’s observation that Proposition 3.1 is ''simple but elegant'' and that Theorem 3.4 ''is also interesting''. We are particularly grateful that the reviewer identified Proposition 3.1 as the conceptual heart of the paper: indeed, one of our main contributions was to show that the stationarity admits a clean quadrature interpretation, and we are very pleased that this message came across so clearly.
>
> ## Weakness: I would appreciate more intuition behind Proposition 3.3.
> On a high level, Proposition 3.3 says that, as the number of particles $n$ increases, the subspace $\\mathcal{F}\_n$ becomes richer and, in the limit, provides a good approximation to the whole RKHS $\\mathcal{H}$. Here, the quality of approximation is measured by the semi-norm $|f|\_n$ which vanishes as $n\to\infty$. In the context of numerical integration, since the quadrature rule is exact on $\\mathcal{F}\_n$ (Proposition 3.1), this means that as $\\mathcal{F}\_n$ grows, the quadrature error for a general function $f\in\mathcal{H}$ is also expected to be small. We will add this extra intuition in the camera-ready version of the paper.
>
> ## Question: Choice of noise level (\beta_t). Is the claim that Gaussian perturbations regularize the particle objective sufficiently to obtain a PL inequality in expectation along the trajectory?
> Yes, exactly; we thank the reviewer for this observation. The motivation for introducing Gaussian perturbations is to regularize the MMD trajectory so that, although the original particle landscape may be highly non-convex, the perturbed objective satisfies a PL-type inequality in expectation along the trajectory. Moreover, the injected noise helps the dynamics escape bad local minima. This is the key condition that makes the convergence analysis tractable, and related ideas have been used in several prior works [3,42].

---

> > ### Author Rebuttal · Reviewer_w7DC · 2026-03-31
> >
> > Thanks for the clarification. I keep my score as is.

---

### Decision · Program_Chairs · 2026-04-30

**Decision:**

Accept (regular)

**Comment:**

This paper studies stationary MMD points, rather than globally optimal minimum-MMD point sets, and shows that they admit an exact quadrature interpretation on a finite-dimensional function space, can exhibit faster integration error decay under suitable conditions, and can be computed via a noisy MMD particle descent procedure. Reviewers viewed the quadrature perspective and the theoretical novelty as the main strengths of the paper. Some concerns were raised about the non-uniqueness of stationary points, but these issues were largely addressed in the rebuttal. Overall, the consensus was that the paper makes a novel and technically strong theoretical contribution, and we recommend acceptance.